# Explaining Bayesian Neural Networks

**Kirill Bykov**[*]
*BIFOLD; TU Berlin, Berlin, Germany; ATB Potsdam, Potsdam, Germany*
*TU Munich; MCML, Munich, Germany*[†]
*kirill.bykov@tum.de*

**Marina M.-C. Höhne**[*]
*ATB Potsdam; University of Potsdam, Potsdam, Germany*
*TU Berlin; BIFOLD, Berlin, Germany;*
*mhoehne@atb-potsdam.de*

**Adelaida Creosteanu**
*Aignostics, Berlin, Germany*

**Klaus-Robert Müller**
*BIFOLD; TU Berlin, Berlin, Germany; Machine Learning Group, TU Berlin, Berlin, Germany*
*Korea University, Seoul, Korea;*
*Max Planck Institute for Informatics, Saarbrücken, Germany*

**Frederick Klauschen**
*Institute of Pathology, Charité — Universitätsmedizin; BIFOLD, Berlin;*
*Institute of Pathology, LMU Munich; German Cancer Consortium (DKTK), Munich, Germany*

**Shinichi Nakajima**
*BIFOLD; TU Berlin, Berlin, Germany; RIKEN AIP, Tokyo, Japan*

**Marius Kloft**
*Department of Computer Science, RPTU, Kaiserslautern, Germany*

**Reviewed on OpenReview:** *https://openreview.net/forum?id=ZxsR4t3wJd*

## Abstract

To advance the transparency of learning machines such as Deep Neural Networks (DNNs), the field of Explainable AI (XAI) was established to provide interpretations of DNNs' predictions. While different explanation techniques exist, a popular approach is given in the form of attribution maps, which illustrate, given a particular data point, the relevant patterns the model has used for making its prediction. Although Bayesian models such as Bayesian Neural Networks (BNNs) have a limited form of transparency built-in through their prior weight distribution, they lack explanations of their predictions for given instances. In this work, we take a step toward combining these two perspectives by examining how local attributions can be extended to BNNs. Within the Bayesian framework, network weights follow a probability distribution; hence, the standard point explanation extends naturally to an *explanation distribution*. Viewing explanations probabilistically, we aggregate and analyze multiple local attributions drawn from an approximate posterior to explore variability in explanation patterns. The diversity of explanations offers a way to further explore how predictive rationales may vary across posterior samples. Quantitative and qualitative experiments on toy and benchmark data, as well as on a real-world pathology dataset, illustrate that our framework enriches standard explanations with uncertainty information and may support the visualization of explanation stability.

---

[*]Equal contribution
[†]Work performed while at TU Berlin and ATB Potsdam.

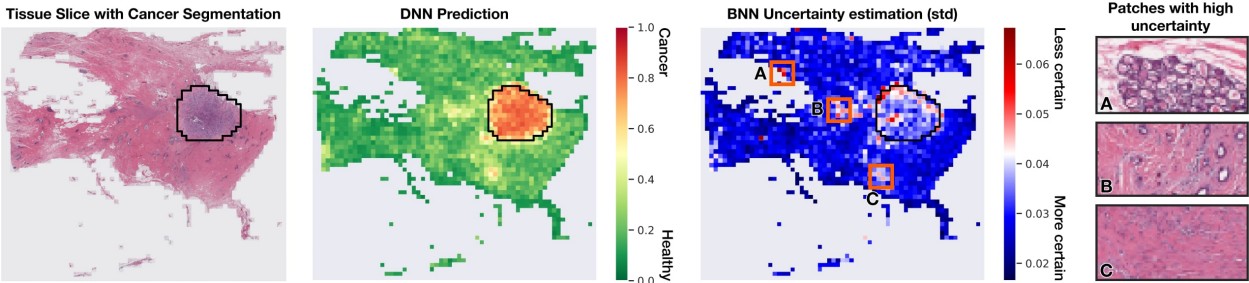

Figure 1: Illustrating the practical advantages of BNNs over standard DNNs. From left to right: a whole-slide image from a cancer patient, divided into patches with highlighted cancerous regions; DNN prediction; BNN uncertainty estimation; and patches with the highest uncertainty. Unlike a standard DNN, the BNN provides uncertainty estimates for each patch. Blue indicates high certainty, while red indicates low certainty. The rightmost panel shows patches where the model's prediction is uncertain and fluctuates between classes.

# 1 Introduction

Deep Neural Networks (DNNs) have achieved significant success over the years, helping to advance Artificial Intelligence (AI). Driven by the exponential growth in available data and computational resources, DNNs achieve state-of-the-art results across various fields of Machine Learning (ML), surpassing human performance in various domains tasks (Silver et al., 2016; Merchant et al., 2023; He et al., 2015; Vinyals et al., 2017; Liu et al., 2019; Won et al., 2020) and, recently, demonstrating emerging general reasoning abilities with the advances of Large Language Models (LLMs) (Bubeck et al., 2023). DNNs accomplish such high performance by learning mappings from raw data to meaningful representations.

With the increasing complexity of modern Neural Networks, it is a difficult task to explain their decision-making process. Therefore, DNNs have often been considered as "black-box" (Vidovic et al., 2015; Buhrmester et al., 2019). However, especially in security-critical applications (such as autonomous driving or medicine), transparency of the decision-making model is mandatory, and therefore, the network's inability to explain its predictions restricts the applicability of ML systems. Indeed, despite showing great performance in test environments, DNNs have not yet reached universal acceptance in the above-mentioned areas (Buch et al., 2018). Furthermore, recent studies have confirmed that frequently DNNs are susceptible to learning *spurious correlations* (Geirhos et al., 2020). These correlations, often arising from chance occurrences or the presence of specific artifacts, are not intended and have undesirable consequences, resulting in poor generalization and potential risks. This phenomenon, termed the *Clever Hans* effect (Lapuschkin et al., 2019) holds significant importance due to its widespread prevalence among various models and datasets (Izmailov et al., 2022). Notably, a substantial number of models trained on the ImageNet dataset (Deng et al., 2009) display reliance on watermarks in Chinese language watermarks (Bykov et al., 2023d), significantly impairing their performance (Li et al., 2023).

The field of *Explainable AI (XAI)* has emerged to address these concerns (see e.g. Samek et al. (2019)). XAI aims to develop and study methodologies for explaining the predictions made by advanced learning machines such as DNNs. Recent advances in XAI have led to a variety of novel methods (Saleem et al., 2022; Adadi & Berrada, 2018; Das & Rad, 2020; Tjoa & Guan, 2020). These can be grouped into *global* and *local* explanation methods. While global explanation methods interpret the decision-making of DNNs across a population by visualizing the signals that cause significant activation in a neuron (Erhan et al., 2009; Nguyen et al., 2016), analyzing relationships between neurons (Bykov et al., 2023a;b) or by linking the neuron's learned concepts to human-understandable concepts (Bykov et al., 2023c; Bau et al., 2017; Mu & Andreas, 2020; Oikarinen & Weng, 2022), local explanations provide interpretations of the prediction for a particular data example by attributing relevances to the input features (Baehrens et al., 2010; Bach et al., 2015; Selvaraju et al., 2017; Yosinski et al., 2015; Kindermans et al., 2017; Montavon et al., 2018; Lundberg & Lee, 2017; Lundberg et al., 2020; Dombrowski et al., 2021).

Most of the explanation methods, local and global ones, are developed for DNNs trained in a *maximum-a-posteriori* (MAP) setting, where the weights of the model are point estimates. In contrast, Bayesian Neural Networks (BNNs) learn a distribution of the network weights, which also induces a distribution on the prediction. For a better intuition on the advantage in explainability of a BNN over a DNN, we consider the example image shown in Figure 1, where a histopathological image (Janowczyk & Madabhushi, 2016; Cruz-Roa et al., 2014) taken from a cancer patient (see e.g. (Klauschen et al., 2018; Hägele et al., 2020; Binder et al., 2021; Klauschen et al., 2024)) is shown on the left. Starting from this whole slide image, smaller patches are extracted and fed into a DNN, trained to classify patches into cancer or non-cancer. In the center of Figure 1, we show the prediction score of a regular (non-Bayesian) DNN for the class cancer. While these scores are usually normalized with a softmax function, these scores do not represent actual probabilities (Hendrycks & Gimpel, 2016) — the DNN provides no further information on how certain or uncertain the patches' relevances are for the prediction. This additional information is provided by BNNs, e.g. in the form of the variances shown on the right in Figure 1. We observe that there are distinct regions where the model is significantly more certain about a patch, indicating cancerous areas (highlighted in blue colors) than in other regions (highlighted in red, referring to the most uncertain regions).

Thus, BNNs provide predictive uncertainty, but—like standard DNNs, do not yield local explanations out of the box. Our contribution is a practical, method-agnostic framework that translates predictive epistemic uncertainty into explanation uncertainty: a distribution over attribution maps that can be summarized by mean and percentile-based aggregations and analyzed via clustering. Within the scope of this work, our goal is to visualize and quantify variability in explanatory rationales under an approximate posterior, without making claims about identifying ground-truth decision modes. Importantly, the computational overhead of our aggregation scales linearly with the number of posterior samples $N$ (i.e., $N$ base attributions plus a constant-time aggregation), and for lightweight approximations such as MC Dropout, the per-sample inference cost differs marginally from deterministic models. Our model is even applicable to regular DNNs (e.g., CNNs) trained in a non-Bayesian fashion, which can first be translated into BNNs (e.g., using MC dropout (Gal & Ghahramani, 2016) or Laplace approximation (De Laplace, 1774; Ritter et al., 2018)), and then using our proposed method for a rigorous explanation. Thus, our approach can enrich the explanation of regular DNNs with additional uncertainty information.

To illustrate this advantage, consider again the medical task illustrated in Fig 1. The first step towards transparency of the shown model is assigning the input features with relevance scores; this is what explanation methods of regular DNNs do. It helps to identify the areas in the image that were relevant for the prediction. Our BNN explanation would additionally provide information on regions where the method is confident about the relevance scores. This may enable an expert to faster identify the most significant cancer areas in an image. On the other hand, the expert might want to look specifically into areas of low confidence to resolve this low confidence by contributing with human expert knowledge to cancer image evaluation. Hence, by determining the level of certainty required by a particular case or by visualizing multiple levels of certainty at once in an explanation, our method could lead to additional insight into the underlying prediction strategy of a model.

In this work, we will propose and investigate different techniques for explaining the decision-making process of (deep) BNNs. Our approach is method-agnostic, i.e., it can build on any arbitrary explanation method for regular (non-Bayesian) DNNs, which are transformed into a local attribution method for BNNs. The proposed method can be combined with any (approximate) inference procedure of BNNs. In computational experiments, our approach interestingly revealed that BNNs implicitly employ multiple heterogeneous prediction strategies. The reason is that BNNs exhibit numerous modes, and approximately one can think of a mode as a prototypical prediction strategy. In contrast, in a standard non-Bayesian DNN, the prediction is deterministic and thus cannot extract multi-modal explanations. With our proposed method, we are now able to visualize the different modes, thus revealing the intrinsic multi-modality in the decision-making of BNNs (and by association: DNNs).

In the following, we summarize the main contributions of this work[1]:

---

[1]Code and reproduction instructions are available at `https://github.com/lapalap/explaining-bnn`.

- **Mean explanation under an approximate posterior:** we provide conditions under which the explanation of the predictive mean equals the mean of explanations (Lemma 3.1, Theorem 3.1).

- **UAI (Union/Intersection) aggregations:** percentile-based summaries that expose stable vs. opportunistic features and yield uncertainty-aware heatmaps (including $UAI^+$).

- **Explanation diversity:** we cluster the sampled explanations in order to visualize diverse explanatory patterns that may arise under the chosen posterior approximation.

- **Generality and cost:** the framework is method- and inference-agnostic and adds only linear overhead $O(N \cdot C_{\text{attr}})$ (with $C_{\text{attr}}$ the cost of the base attribution).

## 2 Background and Related Work

This section provides a comprehensive overview of BNNs and well-established local explanation methods.

### 2.1 Bayesian Neural Networks

From a statistical perspective, standard DNNs are usually trained using maximum a-posteriori (MAP) optimization (Wilson, 2020):

$$\hat{W} = \text{argmax}_W \log p(W \mid \mathcal{D}_{\text{tr}}), \tag{1}$$
$$= \text{argmax}_W \log p(\mathcal{D}_{\text{tr}} \mid W) + \log p(W),$$

which reduces to the maximum likelihood estimation when the prior distribution $p(W)$ is flat. The most commonly used loss functions and regularizers fit into this framework, such as categorical cross-entropy for classification or mean squared error for regression. Although this procedure is efficient since the networks only learn a fixed set of weights, it does not provide uncertainty information about the learned weights and subsequently the prediction. In contrast, Bayesian neural networks (BNNs) estimate the posterior *distribution* of weights, and thus, provide uncertainty information on the prediction, which can provide confidence information on predictions. Particularly, in critical real-world applications of deep learning—for instance, medicine (Hägele et al., 2020; Holzinger et al., 2019; Klauschen et al., 2018) and autonomous driving (Koo et al., 2015; Wiegand et al., 2019)—where predictions need to be highly precise and wrong predictions could easily be fatal, the availability of prediction uncertainties can be of fundamental advantage.

Let $f_W : \mathbb{R}^d \to \mathbb{R}^k$ be a feed-forward neural network with the weight parameter $W \in \mathcal{W}$. Given a training dataset $\mathcal{D}_{\text{tr}} = \{x_n, y_n\}_{n=1}^N$, Bayesian learning (approximately) learns the posterior distribution

$$p(W|\mathcal{D}_{\text{tr}}) = \frac{p(\mathcal{D}_{\text{tr}}|W)p(W)}{\int_{\mathcal{W}} p(\mathcal{D}_{\text{tr}}|W)p(W)dW} \, , \tag{2}$$

where $p(W)$ is the prior distribution of the weight parameter. After training, the output for a given test sample $x$ is predicted by the distribution:

$$p(y|x, \mathcal{D}_{\text{tr}}) = \int_{\mathcal{W}} p(y|f_W(x))p(W|\mathcal{D}_{\text{tr}})dW. \tag{3}$$

Since the denominator of the posterior, shown in equation 2, is intractable for neural networks, numerous approximation methods have been proposed, e.g., Laplace approximation (Ritter et al., 2018), Variational Inference (Graves, 2011; Osawa et al., 2019), MC dropout (Gal & Ghahramani, 2016), Variational Dropout (Kingma et al., 2015; Molchanov et al., 2017), MCMC sampling (Wenzel et al., 2020), and SWAG (Maddox et al., 2019; Wilson & Izmailov, 2020). With these approximation methods, one can now efficiently draw samples from the approximate posterior distribution of the network parameters (equation 3), and compute statistics, e.g., mean and variance, of the prediction for a given data point $x$. Classical MAP training procedures could also be seen as performing approximate Bayesian inference, using the approximate posterior $p(W|\mathcal{D}_{\text{tr}}) \approx \delta(W = \hat{W})$, where $\delta$ is the Dirac delta function.

## 2.2 Local attribution methods

Local explanation methods attribute *relevance* to the input (features) or intermediate nodes (Bach et al., 2015; Vidovic et al., 2016; Selvaraju et al., 2017; Montavon et al., 2018) by using a relevance attribution operation, which we define as follows:

**Definition 2.1** (Relevance Attribution operator). An operator $\mathcal{T}_{x,W}[\cdot]$ that maps an output function $f_W : \mathbb{R}^d \to \mathbb{R}^k$ to a relevance function $R : \mathbb{R}^d \to \mathbb{R}^d$ is called a relevance attribution operator:

$$R_W(x) = \mathcal{T}_{x,W}[f_W](x). \tag{4}$$

The above definition postulates that the relevance of an input feature/node depends on the input mainly via the output function, although it can have a direct dependence on $x$ and $W$.

In this paper, we demonstrate our novel BNN explanation framework mainly using Layer-wise Relevance Propagation (LRP) (Bach et al., 2015) as the base explanation method, however, we would like to stress that our BNN explanation framework can be applied for *any* existing explanation method.

**Gradient explanation**   The Gradient explanation method, i.e. $\mathcal{T}_{x,W} = \nabla_x$, where $\nabla_x$ is the weak derivative w.r.t. $x$, is one of the most basic explanation methods. It visualizes the possible extent of change made by the predictive function in a local neighborhood around the original datapoint $\boldsymbol{x}$ (Erhan et al., 2009; Baehrens et al., 2010; Simonyan et al., 2013).

**LRP**   Layer-wise Relevance Propagation (Bach et al., 2015) is a model-aware explanation technique that can be applied for feed-forward neural networks and can be used for different types of inputs, such as images, videos, or text (Anders et al., 2019; Arras et al., 2017; Montavon et al., 2018; Binder et al., 2021). The underlying idea of the LRP algorithm is to use the network weights and the neural activations computed in the forward-pass to propagate the relevant output back through the network until the input layer is reached. This propagation procedure is subject to a conservation rule — analogous to Kirchhoff's conservation laws in electrical circuits (Montavon et al., 2019) — in each backpropagation step, the relevances from the output layer are distributed towards the input layer, while the sum of relevances should remain the same. Existing variations of LRP are, e.g., LRP-0, LRP-$\varepsilon$, LRP-$\gamma$, and LRP-CMP (Bach et al., 2015; Montavon et al., 2019).

**Integrated gradients**   Integrated Gradients (Sundararajan et al., 2017) is an axiomatic local explanation algorithm that also addresses the "gradient saturation" issue (Samek et al., 2016). It assigns relevance scores to each feature by approximating the integral of the gradients of the model output with respect to a scaled version of the input. The relevance attribution function, in this case, can be defined as

$$\mathcal{T}_{x,W}[f_W](x) = (x - \bar{x}) \odot \int_0^1 \frac{\partial f_W(\bar{x} + \alpha(x - \bar{x}))}{\partial x} d\alpha,$$

where $\odot$ denotes the element-wise product, and $\bar{x}$ is a *reference point* that represents the absence of a feature in the input.

## 3 Explaining Bayesian Neural Networks

**Relation to prior work.**   Work on interpreting Bayesian neural networks (BNNs) has largely focused on predictive uncertainty rather than on distributions of explanations. Kwon et al. (2020) decomposes moment-based predictive uncertainty into aleatoric and epistemic components, and Chai (2018) proposes a model-agnostic approach to visualize the contributions of individual features to predictive, epistemic, and aleatoric uncertainty. These methods analyze uncertainty of the prediction, not the variability of input-level explanations induced by the BNN posterior. Noise-based smoothing methods for deterministic networks take a different route. Bykov et al. (2021) injects multiplicative Gaussian noise into weights to reduce gradient shattering (cf. Samek et al. (2021)) in the spirit of SmoothGrad (Smilkov et al., 2017). While such stochasticity can yield smoother saliency maps, it is not tied to a Bayesian posterior and does not

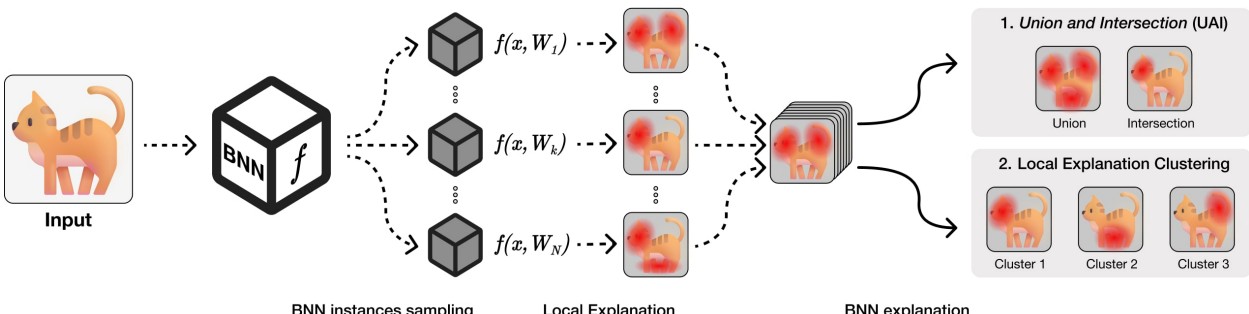

Figure 2: Schematic illustration of proposed methods for explaining Bayesian Neural Networks. Given a particular input – a cat image – we sample models from the posterior distribution and collect local explanations from each instance. These explanations are later aggregated using *Union and Intersection* (UAI) method: The Union explanation provides a global overview of the features learned by the BNN by combining various modes, whereas the intersection explanation provides the intersection strategy used by the BNN – demonstrating the most certain features used for the prediction. Further, local explanations can be clustered to illustrate the different modalities of the decision-making strategies.

provide posterior-consistent distributional summaries of explanations. A complementary line of work explores variability at the representation level. Grinwald et al. (2023) visualizes diversity in latent representations learned by BNNs on the global level using the Feature Visualization method Olah et al. (2018). In contrast, we operate at the input-attribution level, producing per-input distributions over local explanations. We also note domain-specific applications of BNN explainability, e.g., ocean dynamics (Clare et al., 2022), and uncertainty-aware XAI for deterministic models (Marx et al., 2023).

Our work complements these directions in three ways: (i) it is method-agnostic across local attribution techniques and inference schemes; (ii) it operates at the input-attribution level, rather than on latent representations—yielding a distribution of explanations; and (iii) it introduces percentile-based aggregations and explanation clustering as practical tools for summarizing and visually inspecting variability under an approximate posterior. (i) We treat explanations as random variables under the approximate BNN posterior and summarize them with percentiles (Intersection for high agreement; Union for high coverage) and UAI+ for stability-weighted positive relevance. (ii) The framework is method-agnostic across local attribution techniques (e.g., LRP, IG) and inference schemes (ensembles, Laplace, MC-Dropout). (iii) We provide an exploratory pipeline for explaining multi-modality via clustering of per-sample attributions. Concretely, unlike Kwon et al. (2020); Chai (2018), we quantify and visualize *variability of explanations* (not only predictive uncertainty); unlike Bykov et al. (2021), our variability arises from *posterior sampling* rather than externally injected noise; and unlike Grinwald et al. (2023), our summaries are *input-level* and immediately usable for stability/coverage diagnostics.

### 3.1 Method

In the following, we consider a neural network $f_W(x)$ and a relevance function $R_W(x)$ (defined in equation 4), using an arbitrary explanation method. Note that $R_W(x)$ is a deterministic mapping for a fixed parameter $W$. Therefore, the posterior distribution of parameter $W$ induces a distribution over the relevances, resulting in a distribution over relevance maps. Given the posterior distribution of $W \sim p(W|\mathcal{D}_{\mathrm{tr}})$, we can define the distribution of relevance as

$$p(R|\boldsymbol{x}, \mathcal{D}_{\mathrm{tr}}) = \int R_W(x) p(W|\mathcal{D}_{\mathrm{tr}}) dW. \tag{5}$$

The relevance samples

$$R \sim p(R|\boldsymbol{x}, \mathcal{D}_{\mathrm{tr}})$$

can be obtained by drawing weights from the posterior distribution:

$$W \sim p(W|\mathcal{D}_{\mathrm{tr}}).$$

A schematic illustration of the process of obtaining the explanations, as well as a high-level overview of proposed methods, is given in Figure 2. Here, illustratively, we can observe that different samples of the network lead to different explanations of the same input image. Applying aggregation strategies, such as the Intersection, Average, and Union, provides different perspectives on model behavior and may offer additional insights into how predictions are formed.

### 3.1.1 Average Explanation

For some conditions, the average relevance attribution coincides with the relevance attribution of the average prediction, which we state in the following Lemma:

**Lemma 3.1.** *For any explanation method that can be formalized as in equation 4 with a linear operator $\mathcal{T}_{x,W} = \mathcal{T}_x$ that does not depend on $W$, it holds that*

$$\mathcal{T}_x \left[ \mathbb{E}_W \left[ f_W \right] \right] (x) = \mathbb{E}_W \left[ \mathcal{T}_x [f_W](x) \right] = \mathbb{E}_W \left[ R_W(x) \right]. \tag{6}$$

The claim is held trivially by the linearity assumption. Equation 6 in the above Lemma holds for some existing explanation methods, including LRP-0, which is known to be expressed as equation 4 with $\mathcal{T}_{x,W}[f_W](x) = \mathcal{T}_x[f_W](x) = x \odot \nabla_x[f_W](x)$. One can still rely on equation 4 under a slight violation of linearity.

**Theorem 3.1.** *Equation 6 holds almost everywhere in $x \in \mathbb{R}^d$ for LRP-0 with ReLU networks, gradient explanation, and IG.*

The proof is given in the Appendix. Theorem 3.1 states that the explanation of the predictive mean (LHS of equation 6) can be computed by the sample mean of the relevance maps over the posterior distribution (RHS of equation 6).

This is beneficial since it is computationally exhausting to explain the approximate mean predictive function directly (this requires simultaneously storing all the parameters along with their computational graphs for each of the samples from the posterior distribution), using the results from Theorem 3.1 we can now easily explain it by sampling relevance maps, which drastically eases the process.

### 3.1.2 Exploring multi-modality of explanations

As a result of the non-linear network activations, BNNs are known to have multi-modal predictive functions (Bishop, 2006). Different parameters sampled from an approximate posterior can yield noticeably different local attributions for a fixed input (Fig. 2). Consequently, naive averaging can obscure meaningful variability. We therefore cluster-sampled explanations to provide an overview of different explanatory patterns within the posterior. Note that we do not assume that clusters correspond to distinct posterior modes; rather, they offer an exploratory perspective on potential explanation diversity.

To investigate the "prime" strategies of the Bayesian learning machine and to decompose the behavior of BNNs into groups of inter-similar strategies, we propose to cluster the sampled explanations. Although our method does not restrict a user in choosing an algorithm for clustering, we propose to use the SpRAy (Spectral Relevance Analysis) clustering method. This method was initially introduced in (Lapuschkin et al., 2019) to solve a similar task — an analysis of the class-wise learned decision strategies of DNN in order to obtain a global view on the class related relevant patterns, which also supports the identification of undesirable behavior, such as Clever Hans artifacts (see Section 5.4 for more information about Clever Hans behavior). Note that SpRAy was originally constructed to investigate the typical traits in the decision-making process over a large collection of relevance maps, which were obtained from different data points from the training dataset. In contrast to the original setting of SpRAy, we exhibit and identify typical as well as atypical behavior in the BNN decision-making process for a *single* input image.

Once the clustering has been performed, "prime" prediction strategies of the BNN for the given input image can be identified by the cluster-wise average explanations. Moreover, the number of saliency maps in each cluster (normalized by the number of sampled relevance maps) could be considered as the "strength" of each strategy. We could visualize the explanations, clusters, and average cluster strategies in a two-dimensional embedding using a t-SNE plot (Maaten & Hinton, 2008) as shown in Figure 3, where the explanations of

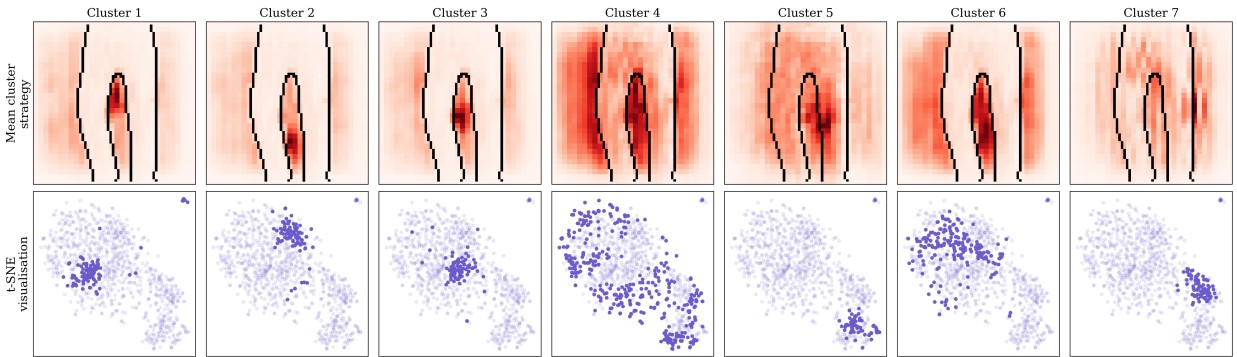

Figure 3: Visualization of the multi-modality of gradient explanations of a BNN (here, a LeNet network trained with dropout) is shown exemplarily for an image of class "Trousers" from the Fashion MNIST dataset. The explanations were clustered by the SpRAy algorithm into 7 clusters, stated on top, and the first row shows the mean explanation for each cluster respectively, where the shape of the trousers is overlaid over the explanation. The second row depicts the t-SNE visualization of the distribution of explanations, where the points of the particular clusters are highlighted. From the mean cluster explanations, we can observe the variability in the decision-making process of the Bayesian Neural Networks — each mode illustrates one decision-making pattern, and the number of elements in each cluster indicates the importance of each cluster to the prediction.

the Fashion MNIST (Xiao et al., 2017) input image is mainly divided into seven different clusters, which indicates the patterns of the main modes of the BNN. More practical details about the clustering process can be found in the appendix.

### 3.1.3 Union and Intersection Explanation

Grasping the multimodality of the network through explanations can be done in several ways. Each relevance attribution map is an explanation for an individual instance of the Bayesian Neural Network. In our work, we observed that differences between instances of BNN are reflected in the multi-modal distribution of explanations. Therefore, to aggregate differences in explanations for Bayesian Neural Networks, we propose a method called **UAI**: Union and Intersection. The intuitive idea is illustrated in Figure 2. We treat the relevance of a BNN as a random variable that follows equation 5.

Given this distribution of relevance maps $p(R|\boldsymbol{x}, \mathcal{D}_{\mathrm{tr}})$, we capture the uncertainty information of the BNN as follows:

$$\mathrm{UAI}_\alpha(x) = \mathcal{P}_\alpha \left[ p(R|\boldsymbol{x}, \mathcal{D}_{\mathrm{tr}}) \right], \tag{7}$$

where $\mathcal{P}_\alpha$ is an operator computing the entry-wise (e.g. pixel-wise in the case of images) percentiles.

High percentiles ($\alpha > 0.5$) correspond to what we introduce as Union explanation – a resulting relevance map consisting of an accumulation of features of various relevance maps, i.e., providing information across modes of the BNN by allocating relevance to features that were considered of high importance by at least a small proportion of samples. In contrast to the Union explanation, small percentiles ($\alpha < 0.5$) illustrate the intersection of features, where at least 95% of explanations agree. For visualisations, we define Union explanations with $\alpha = 0.95$ and Intersection explanations with $\alpha = 0.05$

Furthermore, we introduce $\mathrm{UAI}^+$, as the uncertainty information of the BNN relating to positive class attributions only:

$$\mathrm{UAI}^+(x) = \mathcal{F}_\epsilon \left[ p(R|\boldsymbol{x}, \mathcal{D}_{\mathrm{tr}}) \right], \tag{8}$$

where $\mathcal{F}_\epsilon$ is an operator computing the entry-wise (pixel-wise) probabilities of relevance attributed to a particular pixel being less than some small predefined value $\epsilon > 0$. All proposed approaches (Intersection,

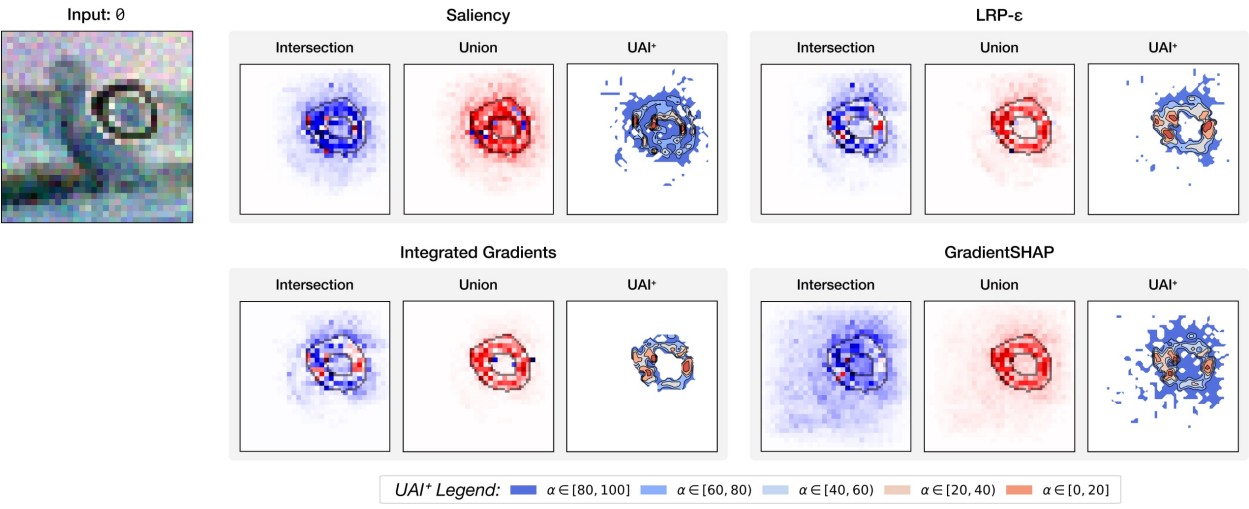

Figure 4: Illustrative explanations of a Bayesian Neural Network. The BNN was trained with Dropout on the Custom MNIST dataset (Bykov et al., 2021). The input, an MNIST digit zero on a random CIFAR Background, is shown on the left and was correctly classified as zero by the BNN. Explanations of the BNN decision are given as Intersection ($\alpha = 5$), Average, Union ($\alpha = 95$), and UAI$^+$ explanations using LRP-$\varepsilon$ (first row), Integrated Gradient (IG) (second row). We can observe that the relevance of all explanations correctly emphasizes the digit. However, our proposed Union and Intersection approach, in which the bundled information is contained in the UAI$^+$ explanation, has a stronger informational content about the role of the features concerning the model prediction by specifying the model's (un)certainty that a feature contributed to the prediction made.

Average, Union, and UAI$^+$) are method-agnostic, and the resulting LRP-$\epsilon$ and IG explanations are illustrated in Figure 4 for the case of an MC Dropout network.

As scales of various explanation methods differ, instead of thresholding significant positive relevances by fixing $\epsilon$, we perform a *group normalization*: we normalize all positive relevances in all sampled attributions $R_i$ by the maximum relevance value:

$$R_i^* = \frac{R_i}{\max\left(r_i^j | r_i^j \in R_i \ \forall i \in [1, N] \ \forall j \in [1, d]\right)}, \tag{9}$$

where $N$ is the number of sampled attributions from the posterior and $d$ is the number of features of the input. This way, we can set $\epsilon$ to a small value on the scale of [0,1], such that it will threshold only significant positive relevances. In our visualisations, we set $\epsilon = 0.05$ as we empirically observe this value to be the borderline of visual recognition of positive relevances in the attribution map.

### 3.1.4 Computational complexity

Let $C_{\text{attr}}$ denote the cost of one base attribution on a deterministic network. Our framework draws $N$ posterior samples and computes $N$ attributions, then applies constant-time, entry-wise aggregations (percentiles or averages). Thus, the total cost is

$$\underbrace{O(N \cdot C_{\text{attr}})}_{\text{attributions}} + \underbrace{O(d\,N \log N)}_{\text{percentiles (optional)}} ,$$

where $d$ is the number of input features (e.g., pixels). For MC Dropout, each forward pass adds only Bernoulli mask sampling and element-wise masking, so $C_{\text{attr}}$ is close to the deterministic cost. The choice of posterior approximation is orthogonal to our method; more expensive samplers (e.g., HMC) increase $C_{\text{attr}}$ proportionally.

# 4 Evaluation procedure

In the following, we provide the methodology of the qualitative and quantitative evaluation.

## 4.1 Qualitative Evaluation

For visual inspection of the results, we normalize the relevance maps with the MinMax transformation (Bach et al., 2015), which maps positive relevances onto the interval $[0,1]$ and negative ones to $[-1,0]$. Afterwards, the normalized relevance maps are visualized using the *'seismic'* colormap[2], which attributes red tones to pixels with positive relevances and blue tones to pixels with negative relevances.

## 4.2 Quantitative evaluation

For quantitative evaluation, we use the localization criterion, where we are interested in measuring the ability of an explanation method to attribute positive relevance to the object of interest. Hence, exemplary, if the prediction of a model is "cat", we assume that in the given image, parts of the object cat are responsible for the prediction and subsequently yield positive relevance attribution by the explanation method. Thus, in order to correctly measure the ability of the method to "find" the object of interest, ground-truth segmentations are required (Arras et al., 2020; Lapuschkin et al., 2016; Kohlbrenner et al., 2020; Zhang et al., 2018; Fawcett, 2006; Bykov et al., 2021).

To measure the localization capability of an explanation method, we employ two different metrics, Area Under the ROC Curve and Relevance Mass Accuracy (Arras et al., 2020).

- **AUC ROC**: For each explanation (e.g., in the form of a heatmap), we calculate the area under the receiver operating characteristic in order to measure how closely the areas of greatest relevance of the explanation correspond to the classified object. Note that for computing the AUC value, the pixel-wise ground-truth segmentation of the object serves as the true label information, whereas the relevance information from the explanation serves as the predicted label information.

- **Relevance Mass Accuracy (MA)**: For each explanation, we measure the proportion of the relevance mass that lies on the object in comparison to the total relevance mass:

$$MA = \frac{\sum_{i \in O} R_i}{\sum_{j \in I} R_j},$$

where $I$ is the set containing all features and $O \subset I$ is a subset of features that are part of the segmented object itself.

While the AUC metric can be used both for explanation methods attributing positive and negative values, as well as for methods attributing just positive relevances, MA can only be used for positive relevances from the explanation method. We acknowledge alternative metrics like Fidelity (Samek et al., 2016), ROAR (Hooker et al., 2019), and F-Fidelity (Zheng et al., 2024), which assess explanation quality via input perturbation. However, these are designed for deterministic models with a single explanation per prediction. Our method produces a distribution over explanations, making direct application of such metrics non-trivial. We therefore use object-localization metrics (AUC-ROC and MA), which are more naturally suited to our probabilistic setting. Extending fidelity-based metrics to BNNs remains an important direction for future work.

## 4.3 Baseline

In our experiments, we compare the proposed UAI explanations with the baseline explanation, which uses the expected value of the weights $\mathbb{E}[W]$. In practice, this would correspond to a MAP classifier, e.g., Laplace approximation or MC Dropout, where the mean weights of the model are used for prediction. Hence, we refer to the baseline explanation as the one explaining the standard deterministic model using the mean

---

[2]https://matplotlib.org/3.1.0/tutorials/colors/colormaps.html

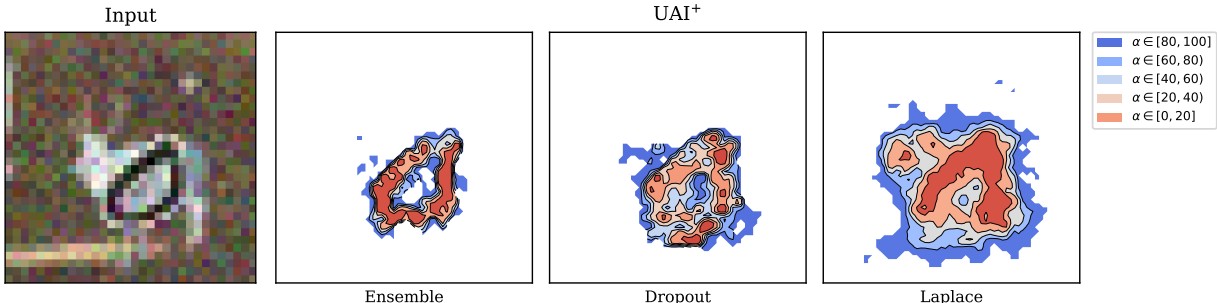

Figure 5: Illustration of UAI$^+$ method explanations based on the Absolute Gradient explanation method for three different Bayesian scenarios: Deep Ensemble, MC Dropout, and Laplace approximation. From the explanations, we can observe the main features that each of the Bayesian Networks used.

weights — this allows us to draw conclusions regarding how standard explanation methods can be enhanced by "Bayesianization".

## 5 Experiments

In the following, we demonstrate the performance of our proposed method both qualitatively and quantitatively.

### 5.1 Experiment on Custom MNIST data

We evaluate our proposed methods on the Custom MNIST (CMNIST) dataset (Bykov et al., 2021), where MNIST digits are plotted on a randomly chosen CIFAR background. This ensures that the decision-making basis for the model should rely only on the number and not on the randomly chosen background. Hence, using the available segmentations of the MNIST digits as a ground-truth indicator of the relevant area the evidence should be distributed to, we are able to measure the goodness of the different explanations. To show that our method is applicable to different types of Bayesian Neural Networks and Network Ensembles, we analyze three different settings, all based on the same standard LeNet architecture (LeCun et al., 1998). We employ three different Bayesian approximation methods — Deep Ensemble of 100 different networks, trained with a random initialization respectively, Laplace approximation, and MC Dropout (more details about the model architecture and training parameters can be found in the appendix). Figure 5 illustrates the differences of UAI$^+$ explanations between the three Bayesian scenarios. For each of the three described scenarios, we used a test set of 10000 generated images, which was not used during training. For each image, $N = 100$ relevances were sampled using the LRP-$\varepsilon$ method from the posterior distribution (in the case of a deep ensemble, each relevance came from a different network instance).

The quantitative results of the localization evaluation for all three scenarios are summarized in Table 1. From the results, we can observe that the Union method is best-performing in terms of AUC metric, while the Intersection method shows overwhelmingly best results in terms of the Relevance Mass Accuracy metric. From these results, we conclude that the Union method indeed is better in visualizing all the information the Bayesian Network has learned about the object; however, comparatively low MA scores of the Union method imply that explanations attribute positive relevance outside of the object of interest. In comparison, the Intersection method has low AUC scores in almost all scenarios, which implies that the Intersection method does not "cover" the object in interest with positive attributions, but high MA scores show high confidence in positive features — if the Intersection method attributes a positive relevance to a feature, it is most likely to lie inside of object of interest.

Table 1: Quantitative results for Localisation (AUC) and Relevance Mass Accuracy (MA). Values are mean $\pm$ s.d. rounded to two decimals.

| | Ensemble | | Laplace Approx. | | Dropout | |
|---|---|---|---|---|---|---|
| | AUC | MA | AUC | MA | AUC | MA |
| **Random** | $0.50 \pm 0.02$ | $0.25 \pm 0.01$ | $0.50 \pm 0.02$ | $0.25 \pm 0.01$ | $0.50 \pm 0.02$ | $0.25 \pm 0.01$ |
| **Baseline** | — | — | $0.54 \pm 0.03$ | $0.85 \pm 0.10$ | $0.53 \pm 0.03$ | $0.92 \pm 0.07$ |
| **Average** | $0.50 \pm 0.05$ | $0.94 \pm 0.05$ | $0.50 \pm 0.06$ | $\mathbf{0.85 \pm 0.09}$ | $0.51 \pm 0.04$ | $0.91 \pm 0.07$ |
| **Intersection** ($\alpha = 5$) | $0.16 \pm 0.06$ | $\mathbf{1.00 \pm 0.00}$ | $0.15 \pm 0.07$ | $0.78 \pm 0.41$ | $0.17 \pm 0.06$ | $\mathbf{0.99 \pm 0.03}$ |
| **Union** ($\alpha = 95$) | $\mathbf{0.88 \pm 0.05}$ | $0.77 \pm 0.09$ | $\mathbf{0.86 \pm 0.07}$ | $0.70 \pm 0.13$ | $\mathbf{0.86 \pm 0.05}$ | $0.87 \pm 0.08$ |
| **UAI$^+$** | $0.79 \pm 0.06$ | $0.90 \pm 0.08$ | $0.78 \pm 0.06$ | $0.79 \pm 0.14$ | $0.71 \pm 0.05$ | $0.94 \pm 0.07$ |

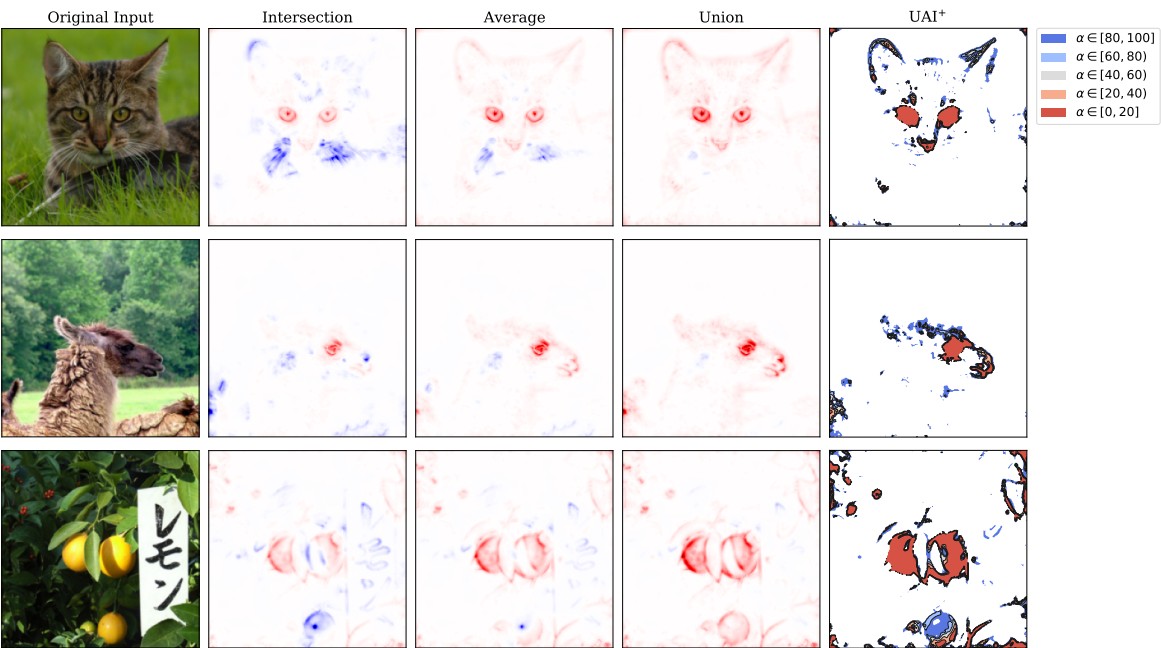

Figure 6: Exemplary explanations of three images taken from ImageNet (red/blue indicates positive/negative relevance). Each row corresponds to a particular input image. From left to right: original image, intersection explanation, baseline (average LRP) explanation, and Union explanations, as well as the UAI$^+$ explanation. We observe that the baseline explanation attributes positive relevance to almost the whole cat except the whiskers. The intersection explanation highlights solely the eyes and the nose of the cat as coherent features whereas the union explanation highlights the same features as the baseline explanation but additionally attributes relevance to the whiskers. The explanation of UAI$^+$ provides a holistic representation of the different feature importance regarding the model decision: red=high importance, (orange, grey, light blue)=intermediate importance, and dark blue=low importance.

## 5.2 Experiment on ImageNet

We demonstrate the usefulness of the proposed method for explaining the decision-making process for a naturally non-Bayesian model that was trained with dropout regularization. We used the broadly applied pre-trained VGG16 network (Simonyan & Zisserman, 2014). This network was pre-trained on ImageNet (Russakovsky et al., 2015) and is naturally non-Bayesian. To access the uncertainty of this naturally non-Bayesian model, we employed MC Dropout (Gal & Ghahramani, 2016) during the test phase, which can always be applied when a model's architecture comprises at least one dropout layer. Furthermore, as a relevance attribution function, we used the LRP-CMP rule as explainability method. For the evaluation of the performance of the proposed methods, we randomly choose a small subset of classes from the ImageNet dataset, consisting

of 3 classes: "tiger cat", "llama", and "lemon". For each class, we downloaded[3] 1000 random images. The explanation results for the Intersection, Average, Union, and UAI$^+$ explanations for three randomly chosen images are shown in Figure 6.

### 5.3 Use case: Prioritized and Coverage-Oriented Explanations (medical data)

In this use case, we study binary classification of histopathological images into cancer vs. non-cancer. In medical imaging practice, clinicians need not only accurate predictions but also actionable, uncertainty-aware rationales that support *prioritized review* of the most reliable evidence and *coverage-oriented sweeps* that reduce the risk of overlooking small foci (cf. Klauschen et al. (2018); Hägele et al. (2020); Binder et al. (2021)). Our UAI method summarizes agreement and coverage of explanatory features across posterior samples, yielding (i) a **prioritized** view (stable cores) and (ii) a **coverage-oriented** view (sensitive, broader evidence).

The dataset comprises 22,302 patches from anonymized non-small-cell lung cancer (NSCLC) cases ($n$=200) drawn from routine diagnostics at the Institute of Pathology, Charité—Universitätsmedizin Berlin, digitized using a 3DHistech P1000 whole-slide scanner. Expert pathologists annotated 28 morphological classes. For this use case, we reduce to binary carcinoma vs. non-cancer. Data are split into training and test partitions. The training set contains 17,884 labeled patches, with 69.9% non-cancer. Point-level annotations of cancer cells are available for a subset of cancer images and are used only for localization evaluation (below).

For this experiment, we trained a VGG-16 (Simonyan & Zisserman, 2014) network with an additional dropout regularization layer applied in the feature extractor part of the network (after the first and third MaxPool layer). The network was trained in a binary classification fashion for detecting the existence of cancer tissue in a histological slide. We trained the network for 100 epochs, using the cross-entropy loss with stochastic gradient descent, where the initial learning rate was set to 0.001. The trained network achieved an accuracy of 86.96% on the provided test dataset and an F1 score of 0.8205. Afterward, we computed the UAI explanations for different test images, which allows us to provide an additional estimate of the explanation uncertainty. Results for three different prototypical cancer images are shown in Figure 7. The original histopathological image is shown in the left column with the black dots representing expert-labeled cancerous cells.

In the right column of Figure 7, the UAI results are plotted over the original image and highlight the relevant parts of the image regarding their importance for the classifier. In the Intersection explanation (second column from the left), we show the regions in the image where the classifier is most certain about their relevance to the prediction "cancer", and with gray color, features that are absent from the Average explanation are highlighted. Analogously, for the Union explanation, we highlight features in green that are attributed positively in the Union explanation but not in the Average explanation.

Table 2: Localization of expert-annotated malignant cells (point annotations). We report AUC (mean ± s.d.) for thresholded attribution maps versus cell annotations on cancer-labeled test images with available points. Higher percentiles (Union, large $\alpha$) increase coverage and yield the best AUC; $\alpha$=99 performs best here. Intersection (small $\alpha$) favors stability/precision over coverage.

| Method | | AUC Score |
|---|---|---|
| Baseline | | $0.6534 \pm 0.1705$ |
| Average | | $0.6635 \pm 0.1712$ |
| UAI | $\alpha = 1$ | $0.5960 \pm 0.1733$ |
| | $\alpha = 5$ | $0.6138 \pm 0.1742$ |
| | $\alpha = 25$ | $0.6536 \pm 0.1716$ |
| | $\alpha = 50$ | $0.6818 \pm 0.1711$ |
| | $\alpha = 75$ | $0.7026 \pm 0.1717$ |
| | $\alpha = 95$ | $0.7179 \pm 0.1694$ |
| | $\alpha = 99$ | $\mathbf{0.7201 \pm 0.1680}$ |

Thus, we can visually observe differences between Intersection, Average, and Union explanations — while Intersection explanations provide a user with more "conservative" explanations, the Union method allows us to observe all the features that were considered with positive evidence towards the class in question. From the quantitative results shown in Table 2, we can observe that the higher the percentile value, the larger the AUC score for the localization criteria of the cancerous cells. Note that in the quantitative experiment, we used only the images labeled as cancerous, where the annotations of the cancerous areas were available.

---

[3]To download a subset of the ImageNet dataset, the following library was used: `https://github.com/mf1024/ImageNet-Datasets-Downloader`.

In general, the UAI-based heatmaps with different $\alpha$ values can prove particularly useful with respect to different diagnostic applications. Low $\alpha$ value explanations can help to identify tissue regions with the highest likelihood of cancer. High $\alpha$ percentile explanations may then be used for AI-based screening applications, where it is important not to overlook even the tiniest occurrence of tumor cells in tissue samples. Therefore, combining UAI analyses with low and high $\alpha$ may provide high sensitivity and simultaneously points the pathologists to regions where the machine is most confident about its decision. This additional information will improve diagnostic speed and also reduce the risk of overlooking crucial information in the diagnostic process.

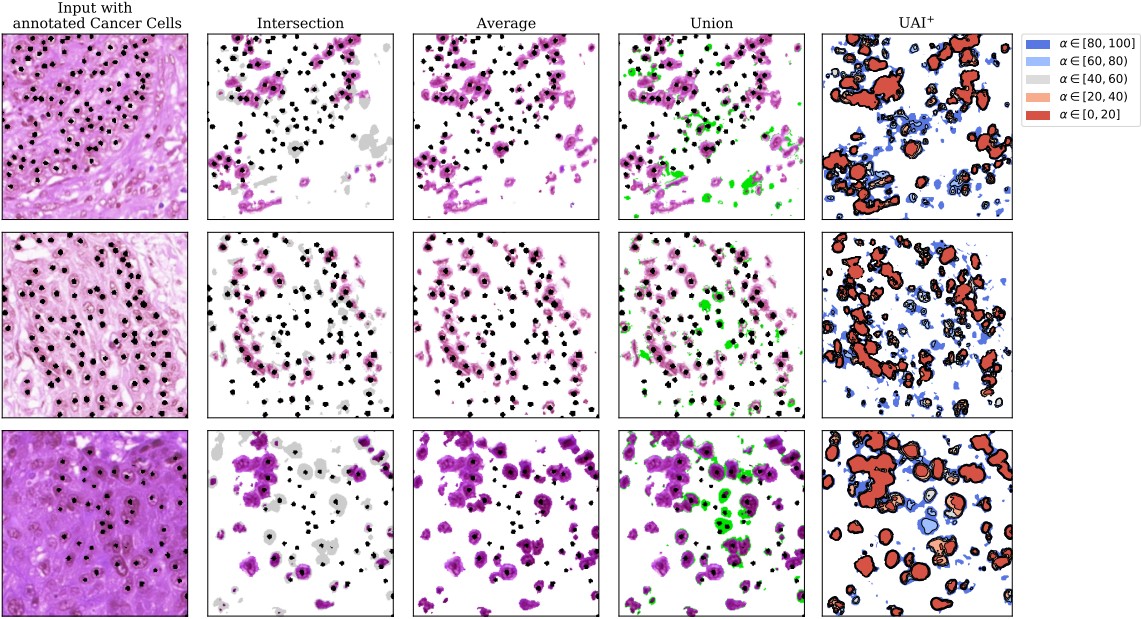

Figure 7: Comparison of the explanation results for the cancer experiment. A VGG16 (Simonyan & Zisserman, 2014) was trained on a set of Haematoxylin-eosin-stained Lung adeno carcinoma (LUAD) as well as on non-cancer histological slides. Three original images labeled as cancer, taken from the test set, are shown on the left, overlaid with black dots, which represent the cancer cells annotated by experts. Behind, the various explanations of the model prediction for the class cancer are shown from left to right as intersection, baseline, union, and UAI$^+$ explanation. For reporting only significant positive relevances, we set a threshold at $\varepsilon = 0.05$ and visualize only relevances that surpass this threshold. We highlight by grey and green the additional information gained by our union and intersection approach in comparison to the baseline explanation. In detail, the intersection highlights the most certain areas, which are indicated as parts of the original image, whereas the areas that are not certain, and thus not visualized in the intersection explanation, are highlighted in grey. In contrast, the union explanation identifies additional information that was not shown in the baseline explanation and therefore may point out new areas that may be cancerous, which are highlighted in green. The aggregation of the diverse levels of feature importance is summarized in the UAI$_+$ explanation shown on the right. Together, these views support a *prioritized review* workflow (Intersection) and a *coverage-oriented* workflow (Union) without retraining the model.

## 5.4 Use case: Confirming the Clever Hans Effect

In this experiment, we revisit the work of Lapuschkin et al. (2019); Samek et al. (2021); Kauffmann et al. (2025) on the *Clever Hans* effect. This term refers to a problematic solution strategy in which a model achieves correct predictions for the wrong reasons. The name originates from a horse named Hans, which appeared to solve arithmetic problems, but was in fact responding to subtle cues from its trainer rather than performing actual calculations.

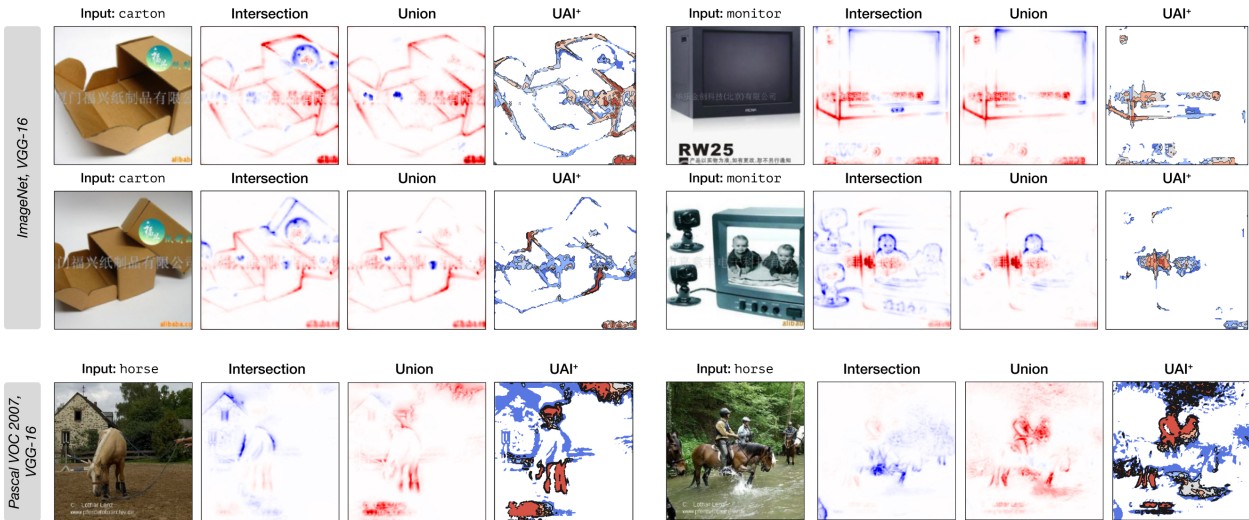

Figure 8: Illustrative visualization of the *Clever Hans* effect in two datasets: ImageNet and Pascal VOC 2007. The UAI explanations assist in distinguishing between random artifacts in the attribution maps and systematic behavior learned by the model. The enhanced method, UAI$^+$, allows us to confirm the Clever Hans effect with high confidence (highlighted in red) in both cases: for ImageNet, where a spurious correlation arises due to a Chinese-language watermark at the center of the image, and for Pascal VOC 2007, where the model relies on photographer watermarks containing name and website (bottom, left part of the images).

In modern machine learning, a *Clever Hans* effect typically involves the model relying on spurious correlations or artifacts—such as watermarks—that are coincidentally associated with specific classes. For instance, an irrelevant visual pattern might consistently appear in training images of one class, leading the model to learn this pattern as a discriminative feature (Lapuschkin et al., 2016; 2019).

We conducted our experiments on two datasets: Pascal VOC 2007 and ImageNet. Prior work (Lapuschkin et al., 2019) has shown that models trained on Pascal VOC 2007 exhibit Clever Hans behavior for the class *horse*, often relying on photographer watermarks. In the case of ImageNet, Bykov et al. (2023d) identified spurious correlations between Chinese-language watermarks and certain classes such as "carton" and "monitor". For both experiments, we used the VGG16 architecture: a model retrained from scratch for Pascal VOC 2007 (see Appendix A.3.3) and a standard pretrained model for ImageNet.

Figure 8 presents the UAI explanations for both datasets. In both cases, watermarks appear prominently in the explanations within the top 5th percentile. This means that for at least 95% of the model samples, the model consistently identifies the watermark as a highly relevant feature for the respective class predictions. Specifically, Chinese watermarks are associated with "carton" and "monitor" in ImageNet, while photographer watermarks are linked to the "horse" class in Pascal VOC 2007.

These findings confirm the presence of the Clever Hans effect: the models rely on artifacts rather than semantically meaningful features. Importantly, our UAI method proves effective in identifying such systematic misbehaviors, helping to differentiate between robust explanations and potential artifacts of the explainability technique itself (see also (Dombrowski et al., 2019; Heo et al., 2019; Samek et al., 2019)).

# 6    Concluding Discussion

When tackling real-world learning problems, Bayesian models have been helpful in assessing the intrinsic uncertainties encountered when predicting. The field of XAI could introduce a further safety layer into the inference process when using neural networks because explanations can contribute to, e.g., unmasking flaws in a model or data set (Lapuschkin et al., 2019). So far, however, no XAI method for Bayesian Neural Networks was conceived.

In this paper, we connect BNNs (with built-in uncertainty quantification) and XAI by proposing a practical, method-agnostic framework for explaining BNNs. As demonstrated, our novel technique (called UAI) is applicable to several popular explanation methods.

By appropriately averaging sampled relevance maps, we can obtain an explanation for the expected predictive function. This allows us to shed light on the decision-making process of BNNs: interestingly, UAI allows us to not only inspect the most relevant pixels for a decision but also their (un)certainties. Our method is thus a formidable starting point for obtaining novel insight into the behavior of Bayesian learning models.

We found that, with a high parameter range of $\alpha$, UAI explains the behavior of the network more informatively, in comparison to a standard mean explanation baseline. By gauging $\alpha$, users can understand the rationale behind a network: with small parameters of $\alpha$, we can observe what features are considered to be contributing towards the prediction regardless of the sampled strategy (intersection explanation). With high parameters $\alpha$, we can understand the features for which at least a small fraction of strategies attribute them with positive relevance (union explanation). Thus, by choosing the parameter $\alpha$, users can choose between more or less risk-averse explanations, depending on the task objective. The proposed UAI explanation additionally helps to understand and reflect the multiple explanation modes inherent in a Bayesian ensemble (cf. Figure 2).

The computational complexity of our presented method is linear in the number of posterior samples. In practice, we found that as few as 100 samples are sufficient to obtain a stable assessment of explanation uncertainty, which keeps the computational cost low. While more samples can be used for finer-grained analysis, the default choice of 100 samples strikes a balance between accuracy and efficiency, which makes our approach feasible even for large-scale applications.

Our contribution has several limitations: (i) conclusions are conditioned on the chosen posterior approximation (e.g., MC Dropout, Laplace) and do not claim recovery of true posterior multi-modality; (ii) fidelity-style metrics are not yet standardized for distributions of explanations; and (iii) a full human-in-the-loop evaluation is beyond the scope of our work.

Concluding, our UAI framework extends existing attribution methods to Bayesian Neural Networks by generating and aggregating distributions of explanations. This enables researchers and practitioners to obtain complementary information about the variability of the model reasoning. Our results suggest that incorporating epistemic uncertainty can adjust explanations toward different desiderata: for example, union aggregations improve sensitivity (AUC-ROC), while intersections emphasize stability (MA). Our work provides a foundation on which future research can build up in several directions. First, expanding evaluation criteria to better capture the properties of uncertainty-aware explanations is a necessary next step. Second, practical deployment scenarios deserve further exploration, particularly in safety-critical domains. For instance, in pathology, Bayesian XAI may reveal both spurious and real evidence as separate clusters, potentially reducing the risk that users overlook true signals when spurious patterns are present. More generally, investigating how explanation diversity can support practitioners in identifying unstable reasoning represents a promising direction toward application-specific deployments.

## Acknowledgements

This work was partly funded by the Federal Ministry of Research, Technology and Space (formerly BMBF) through the projects Explaining 4.0 (ref. 01IS200551), REFRAME (ref. 01IS24073B), PIAD (ref. 01IS24071A), and BIFOLD – Berlin Institute for the Foundations of Learning and Data (ref. BIFOLD24B). KRM was funded in part by the Institute of Information & Communications Technology Planning & Evaluation (IITP) grants funded by the Korea Government (No. 2019-0-00079, Artificial Intelligence Graduate School Program, Korea University). MK acknowledges support by the DFG through FOR 5359 (KL 2698/6-1, KL 2698/7-1), TRR 375 (ID 511263698), SPP 2298 (KL 2698/5-2), and SPP 2331 (KL 2698/11-1), and by the Carl-Zeiss Foundation through the initiatives AI-Care and Process Engineering 4.0.

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

# A Appendix

In the following, we provide the supplementary material to our paper Explaining Bayesian Neural Networks.

## A.1 Proof of Theorem 3.1

*Proof.* For LRP-0, it holds that

$$R_W(x) = \mathcal{T}_x[f_W](x), \tag{10}$$

with $\mathcal{T}_x[f_W](x) = x \odot \nabla_x[f_W](x)$. Furthermore, the linearity

$$\tau_1 \mathcal{T}_x[f_1](x) + \tau_2 \mathcal{T}_x[f_2](x) = \mathcal{T}_x[\tau_1 f_1 + \tau_2 f_2](x)$$

holds for any $\tau_1, \tau_2 \in \mathbb{R}$ and any weakly differentiable functions $f_1, f_2$ except at the non-strongly-differentiable points of $f_1$ and $f_2$. We will prove, that in this case

$$\mathcal{T}_x\left[\mathbb{E}_W\left[f_W\right]\right](x) = \mathbb{E}_W\left[\mathcal{T}_x[f_W](x)\right] \tag{11}$$

holds except for measure zero points.

Let us fix $x$. If $f_W$ is strongly differentiable at $x$ for all $W$ on the support of its posterior distribution $q(W)$, linearity of $\mathcal{T}_x$ holds and therefore Equation 11 holds due to Lemma 1. Let us define

$$\mathcal{W}_x = \{W; f_W \text{ is not strongly differentiable at } x\}, \quad \mathcal{Z} = \{x; \int \delta(W \in \mathcal{W}_x)q(W)dW > 0\},$$

where $\delta(\cdot)$ denotes the Dirac measure. Equation 11 still holds for $x \in \overline{\mathcal{Z}}$, because the contribution from the set of non-differentiable models at $x$ is zero. For any bounded distribution $q(x)$ in the input space, it holds that

$$\int q(x) \int \delta(W \in \mathcal{W}_x)q(W)dW dx = \int q(W) \int q(x)\delta(W \in \mathcal{W}_x)dx dW = 0$$

due to the weakly differentiable assumption on $f_W$. This implies that $\mathcal{Z}$ is a measure zero set, which proves the claim for LRP-0.

Similarly, the gradient explanation, as well as IG, can be written as equation 10 with an operator $\mathcal{T}_x$ linear on all strongly-differentiable-points, and the same discussion applies to proving the claim. $\square$

## A.2 Details of the clustering procedure

For the image classification task we employ the SpRAy algorithm as follows (Lapuschkin et al., 2019):

1. **Relevance maps sampling**

   A collection $\{R_i\}_{i=1}^N$ of relevance maps is sampled from the posterior distribution.

2. **Preprocessing of the relevance maps.**

   All relevance maps are normalized using the MinMax normalization procedure. If needed, all relevance maps should be made uniform in shape and size, for future clustering.

   To speed up the clustering process and to produce more robust results we propose to use downsampling methods on the collection of samples. While the user is free to choose any dimensionality reduction method, we propose to use the Average Pooling method in order to achieve visually different strategies in different clusters.

3. **Spectral Cluster (SC) analysis on pre-processed relevance maps.**

   Pre-processed relevance maps are clustered by Spectral Clustering (SC) method. The affinity matrix, which is necessary for SC method, is based on k-nearest-neighborhood relationships. More detailed,

the affinity matrix $M = (m_{ij})_{i,j=1,...,N}$, measures the similarity $m_{ij} \geq 0$ between all $N$ samples $R_i$ and $R_j$ of a source dataset and is constructed in the following way:

$$m_{ij} = \begin{cases} 1 & \text{if } R_i \text{ is among the k nearest neighbors of } R_j \\ 0 & \text{else} \end{cases}$$

Since this rule is asymmetric, the symmetric affinity matrix $M$ is created by taking $m_{ij} = max(m_{ij}, m_{ji})$. Authors of the original paper highlight that similar clustering results were obtained using the Euclidean distance, with only small differences in the eigenvalue spectra.

The Laplacian $L$ is computed from $M$ as follows :

$$d_i = \sum_j m_{ij}.$$

$$D = diag\,[d_1, d_2, ..., d_N]\,.$$

$$L = D - M.$$

The Matrix $D$ is a diagonal matrix, which describe the measure of connectivity of a particular sample $i$ with $D$ being a diagonal matrix with entries $d_{ii}$ describing the degree (of connectivity) of a sample $i$ (Lapuschkin et al., 2019).

4. **Identification of interesting clusters by eigengap analysis**

By performing an eigenvalue decomposition on the Laplacian $L$, eigenvalues $\lambda_1, \lambda_2, ..., \lambda_N$ are obtained. The number of eigenvalues $\lambda_i = 0$ identifies the number of (completely) disjoint clusters within the analyzed set of data.

The final step of SpRAy assigns cluster labels to the data points, which can then be performed using an (arbitrary) clustering method: in our work we use k-means clustering on the $N$ eigenvectors. The number of clusters can be obtained by *eigengap* analysis (Von Luxburg, 2007): it can be identified by eigenvalues close to zero as opposed to exactly zero, followed by an eigengap — rapid increase in the difference between two eigenvalues in the sequence $|\lambda_{i+1}\lambda_i|$ (Lapuschkin et al., 2019).

## A.3 Experimental setup

### A.3.1 CMNIST experiment

For the CMNIST experiment we used a simple convolutional network similar to LeNet (LeCun et al., 1998). For the Ensemble and for the Laplace scenarios a standard architecture was used, with only a change in number of input channels adjusted to 3-dimensional RGB inputs. For the Dropout scenario, after 2 Average Pooling layers, a 2-d Dropout layer was inserted with a dropout probability set to 0.25. 2 1D Dropout layers were added in the classification part of the network, with the probability of dropout set to 0.5.

All of the networks were trained with a batch size of 32, and with a Stochastic Gradient Descent algorithm, (Léon, 1998) with a learning rate of 0.01 and 0.9 momentum. A learning rate scheduler was used with the number of steps set to 7 and multiplicative parameter $\gamma = 0.1$. For the Ensemble scenario, 100 networks were trained for 20 epochs, and for the Laplace and Dropout, the number of epochs was set to 100. For the Laplace approximation, KFAC Laplace approximation was used (Humt et al., 2020; Lee et al., 2020), with Laplace regularization hyperparameters (additive and multiplicative) both set to 0.1.

### A.3.2 Carcinoma experiment

For the Cancer experiment, we employed a standard VGG-16 (Simonyan et al., 2013) with additional 2D Dropout layers after each MaxPooling layer, with the probability of dropout set to 0.1 and 2 1D

Dropout layers in the classificator part of the network after each activation function with the probability of 0.5. The network was trained with SGD with 0.001 learning rate and 0.9 momentum. `torch.optim.lr_scheduler.ReduceLROnPlateau`learning rate scheduler was used with the factor of 0.1 and patience of 10.

### A.3.3 Clever Hans experiment

For the Pascal VOC 2007 multi-label classification experiment, we employed a standard VGG16 network (Simonyan & Zisserman, 2014), and adjusted the number of output neurons from 1000 to 20, which is the number of different classes in the Pascal VOC 2007 dataset.

We resized each training image, such that the shorter axis has 224 pixels, keeping the aspect ratio unchanged. Then, we randomly cropped the longer axis and obtained square images with the size 224×224. We trained the network for 60 epochs, by minimizing the Binary Cross Entropy loss preceded with a Sigmoid layer[4]. We used the Adam optimizer with its parameters set to $\alpha = 0.0001, \beta_1 = 0.9, \beta_2 = 0.999$. Our trained VGG16 network achieves 91.6%[5] in the multi-label classification on the test set, for which center cropping with square size of 224×224 was applied, instead of random cropping.

### A.3.4 Fashion MNIST experiment

For the Fashion MNIST experiment we trained a LeNet network with 2 2D Dropout layers with $p = 0.5$ added after each of the Average Polling layers in the feature extractor part of the Network, and 1 1D Dropout layer with $p = 0.5$ after the Flatten Layer. The network was trained on FashionMNIST dataset with several augmentations, such as Color Jittering, Random Affine Transformations, and Random Horizontal Flips. The batch size was set to 64, the Network was trained using the Cross-Entropy loss for 50 epochs with Adam optimizer with standard parameters and a learning rate of 0.001.

---

[4]`https://pytorch.org/docs/master/generated/torch.nn.BCEWithLogitsLoss.html`
[5]`https://scikit-learn.org/stable/modules/generated/sklearn.metrics.accuracy_score.html`

