# OpenReview forum: "Explaining Bayesian Neural Networks"
_TMLR — Accepted by TMLR_

### Review · Reviewer_noQ2 · 2025-07-08

**Summary Of Contributions:**

This paper proposes a method-agnostic framework for explaining the decision-making process of Bayesian Neural Networks (BNNs). The approach transforms existing attribution methods for deterministic neural networks into local explanation tools applicable to BNNs by aggregating and analyzing multiple relevance maps sampled from the posterior. Through clustering these explanations, the method reveals the implicit multi-modality in BNN predictions, where different parameter samples may reflect distinct decision strategies. The framework is compatible with various approximate inference techniques, such as Laplace approximation, MC Dropout, and Deep Ensembles.

**Audience:**

Yes

**Claims And Evidence:**

Yes

**Requested Changes:**

1. The central claim that your method reveals the multi-modality of BNN decision-making is compelling but potentially overstated, given that many of the inference techniques used (e.g., Laplace approximation, MC Dropout) are unimodal or cover limited posterior regions. Clarifying the distinction between local explanation variability and genuine multimodal behavior is important to avoid overinterpretation. Please revise the narrative to acknowledge the limitations of the inference methods used and adjust claims about multi-modality accordingly, unless supported by more expressive posterior approximations.
2. The explanation diversity observed in your results may reflect epistemic uncertainty due to weight variability rather than distinct predictive strategies. Clarifying this distinction is essential for supporting your main claim. Consider proposing a metric or analysis—e.g., relating clusters to prediction differences or uncertainty quantification—that distinguishes between these two phenomena.
3. While the clusters of relevance maps are visually distinct, it is unclear whether they correspond to meaningful differences in model behavior (e.g., predicted class, spatial focus, or decision boundary shift). A quantitative evaluation—such as comparing prediction confidence or class probabilities across clusters—would strengthen the argument that these clusters reflect distinct decision strategies.

**Strengths And Weaknesses:**

**Strengths:**
1. Method-agnostic design allows integration with any standard explanation method or BNN inference approach, enhancing flexibility and usability.
2. The use of relevance map clustering provides an intuitive and effective way to explore predictive diversity in BNNs.
3. The experimental evaluation demonstrates broad applicability across different inference methods and datasets.
4. This paper is well-written and well-organized.

**Weaknesses:**
1. The method relies on approximate inference schemes that may not capture the true multimodality of the posterior, especially in the case of unimodal approximations like Laplace.
2. The observed diversity in explanations may result from local perturbations around a single mode rather than genuinely distinct predictive strategies.
3. The approach lacks a principled way to distinguish between epistemic uncertainty and actual mode switching in decision mechanisms.

---

> ### Author Response · Authors · 2025-08-13
> **Rebuttal to Reviewer noQ2**
>
> We thank the reviewer for their insightful and constructive review. We appreciate their recognition of our method’s generality and your thoughtful suggestions regarding the interpretation of explanation diversity.
>
> 1. **Overstatement of multi-modality given limited inference techniques**
> > "Clarifying the distinction between local explanation variability and genuine multimodal behavior is important..."
>
> We agree that the expressive power of posterior approximations (e.g., Laplace, Dropout) limits the extent to which one can capture full posterior multi-modality. We have revised our narrative to more cautiously frame our claims, emphasizing that our method surfaces explanation diversity under a given approximate posterior, rather than making strong claims about full posterior multi-modality.
> Our goal is not to claim ground-truth identification of modes, but rather to provide tools to visualize and quantify epistemic variability in explanations. In this light, explanation clustering acts as a lens for exploring diversity in model reasoning across sampled parameters—even when those come from unimodal or locally concentrated distributions.
>
> 2. **Need to distinguish between epistemic uncertainty and decision-mode switching**
> > "Explanation diversity may reflect epistemic uncertainty... Clarifying this distinction is essential."
>
> We appreciate this important distinction and have clarified it in the revised manuscript. Our current framework does not assume that explanation clusters correspond directly to distinct decision modes. Instead, it visualizes how epistemic uncertainty in weights can translate into variability in input feature relevance, which may or may not imply different predictive strategies.
> While we agree that developing principled criteria for separating explanation variability due to epistemic uncertainty vs. true decision mode switching is an important direction, it lies beyond the scope of the current work. We now clearly acknowledge this limitation and present our results as an empirical tool for exploration rather than a formal decomposition of uncertainty sources.
>
> 3. **Lack of evidence that clusters correspond to different decisions**
> > "It is unclear whether clusters correspond to meaningful differences in model behavior... A quantitative evaluation would strengthen the argument."
>
> We appreciate this point and agree that clustering per se does not guarantee distinct decision strategies. Our goal here is illustrative and diagnostic rather than identificatory: we cluster the outputs of established attribution methods (e.g., IG, LRP) to organize how these methods depict the model’s rationale across posterior samples. In other words, the explanations themselves carry the semantic burden—they are the lens through which the model’s local reasoning is rendered—and clustering simply structures that lens to make variability legible. We therefore refrain from claiming that clusters identify true posterior “modes”; instead, we present them as a compact summary of explanation diversity under an approximate posterior. Within this framing, visually distinct clusters are informative insofar as the underlying explanation method is faithful; when clusters differ, they highlight inputs where epistemic variability manifests as different attribution patterns, and when they do not, they indicate local variability around a stable rationale. We have clarified this scope in the manuscript and explicitly marked clustering as a diagnostic tool to illustrate explanation variability, not a proof of distinct decision mechanisms.

---

### Review · Reviewer_ovyt · 2025-07-23

**Summary Of Contributions:**

The paper proposes the explanation methods for Bayesian neural networks. It claims that while the research focuses mostly on the non-Bayesian deep neural networks, the Bayesian neural networks can provide a new way of interpreting predictions in neural networks.

**Audience:**

Yes

**Broader Impact Concerns:**

No further concerns

**Claims And Evidence:**

Yes

**Requested Changes:**

It would be important to contrast and compare with some of the works addressing a similar challenge such as:
[1] Clare, M. C., Sonnewald, M., Lguensat, R., Deshayes, J., & Balaji, V. (2022). Explainable artificial intelligence for Bayesian neural networks: Toward trustworthy predictions of ocean dynamics. Journal of Advances in Modeling Earth Systems, 14(11), e2022MS003162.

[2] Marx et al (2023) But Are You Sure? An Uncertainty-Aware Perspective on Explainable AI, AISTATS 2023

[3] Grinwald et al (2023) Visualizing the Diversity of Representations Learned by Bayesian Neural Networks, TMLR, 2023

The answers on the questions about Claims and Evidence and Audience are currently No, which is subject to the rebuttal responses.

**Strengths And Weaknesses:**

Strengths:
- Claims and evidence: the maths and the claims about the algorithm performance appear correct, with the authors giving a number of points in support of the claims (theoretical justification, new method, multi-modality of BNNs, generality)
- Audience: the intersection of uncertainty quantification  and interpretability, and BNNs in particular, is undoubtedly of interest

Weaknesses:

Right now, with both of the strengths above there are caveats which I would hope the author can address.

**Claims and evidence:**

It seems that the claims are put from the perspective that the combination of BNNs and interpretability is a novel one. However, the authors need to confirm why these claims are not straightforward combination of BNNs and interpretability. Another TMLR work[3] provides a range of empirical findings about the choice of uncertainty quantification (dropout, posterior approximation),  as well as findings on the link between uncertainty and diversity of features. The theoretical and empirical claims, however, are presented as the initial steps towards BNNs and interpretability, despite that in recent years, these steps have been made. In particular, I wonder if the authors could contrast the differences (and demonstrate their importance) compared to [1]-[2]. In particular, I would expect to see both comparisons in Section 6 and Sections 3-4.

**Audience:**

First, it seems like a number of other papers have already looked into this question from different perspectives (see, e.g. works [1-3]). The audience would wish to find new insights which cannot be obviously concluded  from other papers. In other words, the authors need to clearly emphasise what this work show in a different light form the existing studies of uncertainty quantification.

---

> ### Author Response · Authors · 2025-08-13
> **Rebuttal to Reviewer ovyt**
>
> We thank the reviewer for their detailed feedback and for acknowledging the correctness of our theoretical claims and the relevance of the intersection between BNNs, interpretability, and uncertainty quantification. We respond below to the main points of the reviewer regarding related work and positioning.
> 1. **Novelty of the Contribution and Relation to Prior Work**
> >“It seems that the claims are put from the perspective that the combination of BNNs and interpretability is a novel one… The authors need to confirm why these claims are not a straightforward combination…”
>
> We appreciate this important point and agree that recent works have explored aspects of interpretability in BNNs. Our contribution builds on this growing body of research, but focuses on a method-agnostic, structured framework that explicitly aggregates, analyzes, and clusters multiple local explanations derived from the posterior distribution. While prior works such as [1–3] study uncertainty or explanation variability qualitatively, our method is designed to extract and visualize epistemic diversity in explanations in a systematic, quantitative way, across different BNN types and explanation methods.
> In particular, in contrast to [3], which focuses on feature visualization, a  method for global explanation of neurons, our framework operates at the input attribution level, offering a complementary perspective that makes the diversity in reasoning paths accessible to practitioners.
>
> To our knowledge, no prior work provides a clustering-based approach for systematically characterizing explanation-level diversity across posterior samples in BNNs. We agree that our original framing may have overstated the novelty, and we have revised the manuscript to clearly address this point (Introduction/Conclusion).
>
> 2. **Clarifying the Contribution to the Intended Audience**
> >“The audience would wish to find new insights which cannot be obviously concluded from other papers…”
>
> We have revised the manuscript to more explicitly state what our method reveals that prior work does not. Our contribution lies not in proposing a new explanation method per se, but in leveraging posterior sampling to systematically surface and organize epistemic variability in feature attribution, which is typically hidden in standard XAI workflows.
> The framework equips practitioners with tools to: Visualize explanation variability across samples; Detect possible failure modes or inconsistencies in model reasoning; Identify ambiguous or multi-modal input-output behaviors that may not be obvious from uncertainty in predictions alone.
> We hope this more nuanced positioning will clarify the novelty for the audience and distinguish our work from prior approaches.
> We thank the reviewer again for encouraging us to clarify our claims and better position our work within the literature. We believe the revisions address these concerns and help refine the contribution for the target audience.

---

> > ### Comment · Reviewer_ovyt · 2025-08-21
> >
> > I think the rebuttal convinces me that the contribution has audience, which can be interested in the epistemic variability in feature attribution.
> >
> > On the subject of claims and evidence, taking into account  the other reviews and rebuttals, I unfortunately still have concerns. The authors state in the rebuttal that 'no prior work provides a *clustering-based* approach for systematically characterizing explanation-level diversity across posterior samples in BNNs. ' However, this makes me think about the other concerns raised by the other reviewers, related to the evaluation of the proposed method.
> >
> > In response to Reviewer noQ2 on their suggestion for quantitative evaluation, the authors write: "We appreciate this point and agree that clustering per se does not guarantee distinct decision strategies. Our goal here is illustrative and diagnostic rather than identificatory: we cluster the outputs of established attribution methods (e.g., IG, LRP) to organize how these methods depict the model’s rationale across posterior samples. " But then, if clustering-based approach is a distinguishing part of this work, it is contradictory to the claim that the goal of using clustering is only illustrative.
> >
> > It also comes with the concern from Reviewer Na4v about the utility of the proposed explanation method: what are the use cases for such a method, and why the architectural decisions such as clustering help address these use cases. In other words, for the contributions, listed in the intro, namely *UAI (Union/Intersection) aggregations* and *Explanation diversity as a diagnostic*, what justification would the authors give to the proposed method and what does it contribute towards solving use-case scenarios? Ideally, I would like to see it in the introduction, stating the gap in the existing literature and then, at the end, as a list of claims, where the authors refer in which section they justify the proposed method decisions, including through the use-case evaluation. In the conclusion, I would also like to see the  summary of findings which refer to the quantitative and theoretical arguments.

---

> > > ### Author Response · Authors · 2025-08-25
> > > **Official Comment to Reviewer ovyt**
> > >
> > > > “if clustering-based approach is a distinguishing part of this work, it is contradictory to the claim that the goal of using clustering is only illustrative.”
> > > We thank the reviewer for pointing out this ambiguity and for encouraging us to clarify both the role and the practical motivation of our work. We agree that parts of our original framing could be overstated, and we have revised the manuscript accordingly.
> > >
> > > **The role and novelty of Clustering:**
> > > Our framework combines three components: mean explanations under the posterior, percentile-based UAI (Union/Intersection) explanations, and clustering of posterior attributions. Our clustering approach is novel to the best of our knowledge, since no prior work has systematically applied it at the input attribution level to visualize explanation diversity across posterior samples. At the same time, its role is diagnostic: within our paper clustering does not aim to recover ground-truth decision modes, but to structure and visualize the variability such that users can more easily recognize when different explanatory patterns arise. We have revised the manuscript to make this dual perspective explicit and to avoid any impression of contradiction.
> > >
> > > **Utility and evidence:**
> > > Our aim is not to claim that uncertainty alone solves interpretability challenges, but to show that incorporating epistemic uncertainty might improve established XAI evaluation criteria by revealing additional exploratory information. Basic XAI evaluation methods could not be used out of the box, such as pixel-flipping, since the decrease in model performance over the perturbation of relevant pixels can not be compared directly regarding the different number of relevant pixels for the intersection, mean and union explanation. Therefore we used AUC and mean attribution (MA) metrics, and we find that UAI unions improve AUC, whereas intersections improve MA. These results suggest that uncertainty-aware explanations can be adapted to different practical needs, i.e.,  the union emphasizes sensitivity by capturing a broad set of relevant features, while the intersection emphasizes stability by highlighting only the most consistent ones. Clustering, in turn, provides complementary qualitative insight by organizing diverse explanatory modes, which might support practitioners inspecting potential sources of variability or instability.
> > >
> > > **Future work and use cases:**
> > > While our current contribution focuses on theoretical results and quantitative evaluation, we agree that practical deployment scenarios are of immense importance, especially in safety-critical areas. Therefore, future research will explore potential use cases building on our UAI approach, with particular emphasis on leveraging clustering to surface explanatory diversity in risk-sensitive domains. For example, in pathology, Bayesian XAI may consistently reveal both spurious and real evidence as separate clusters, thereby reducing the risk that users overlook the true signal when spurious patterns are present. Investigating how our framework can further support practitioners in identifying unstable reasoning represents a promising direction toward application-specific deployments. Furthermore, future work will focus on the expansion of XAI evaluation criteria for assessing the quality of uncertainty-aware explanations.

---

> > > > ### Comment · Reviewer_ovyt · 2025-08-28
> > > >
> > > > I think these answers substantially improve the manuscript, and after checking the match between the claims and the evidence I suggest that now the work addresses this criterion. Thank you for making improvements in the paper during the discussion stage.

---

### Review · Reviewer_Na4v · 2025-07-26

**Summary Of Contributions:**

The paper proposes to provide explanations for Bayesian Neural Networks (BNNs).
The main idea is that with linear operators functions (or ReLUs almost everywhere), the explanation of the posterior distribution is simply the distribution of explanations.
This simplifies the story to simply generate a distribution over explanations using standard attribution techniques and then aggregating these individual explanations.
The paper proposes to aggregate based on the percentiles of relevance scores for each pixel (rather than mean or variance of pixel scores) to showcase the variance.
The intersection and union parts are based on setting different values of the percentiles.
Furthermore, the paper claims that this can be applied to standard DNNs that have dropout layers.
The paper shows results on how the various explanations method compare by seeing how much of the object the explanation covers and how much of the non-object it covers.
The paper illustrates on some toy datasets and several realistic datasets.

**Audience:**

Yes

**Claims And Evidence:**

No

**Requested Changes:**

- Give concrete use-cases (likely framed as a decision problem) with concrete users and explain precisely why the uncertainty matters. You should include cases where the explanation properly increases the confidence in the prediction and where it DECREASES the confidence in the prediction appropriately. It must do both, otherwise it could be fooling the user or be completely useless.

- Explain how the metric is a (proxy) measure for simulating the use case that you mention. Why does a better score on this metric imply that your method would perform better on the use case? Further, make sure to include best alternatives like non-Bayesian and possibly other explanation approaches.

**Strengths And Weaknesses:**

**Strengths**
- Paper is well-written and easy to follow overall.

- The histopathology and watermark examples help provide concrete examples over benchmark datasets.

- Understanding uncertainty of explanation is an important facet of explanation in general.

- The basic idea is simple and can be applied to standard DNNs with dropout.

**Weaknesses**
- The paper (like many attribution-based explanation papers) lacks concrete use-cases of when the explanation will actually be useful to someone. There are a fair bit of statements claiming (without direct evidence) that uncertainty in explanation would be helpful for clinicians or practicioners. But in what way? What decision could it actually help?
   - The two concrete use cases seem to be: (1) Histopathology prediction and (2) training data debugging (i.e., watermark). Note that these are two very different use cases with very different users.
   - In the first, the user is the model developer. What other ways could they find "problematic" images or wrong attributions in the images? And how would they actually use this? If they are not given ground-truth segmentation boxes, then how would they use your explanation method? Would they have to look at the explanations for every image? This seems time consiuming and probably too difficult in practice. They could also use explanation methods that select important training samples for the prediction. https://arxiv.org/pdf/2212.04612
   - In the second, the user is the pathologist. In this case, how is showing them more or less information helpful? You need an example where the explanation would confirm the model's prediction AND a case where the explanation would make the pathologist REJECT the model's prediction. If only the first, then it's just reinforcing the bias of the model. You need an example of both cases. AND you need to show that a non-Bayesian approach would fail in one or both of these.

- Relatedly, for those use cases, what are the most natural alternative solutions to the use-case? For example, you would need to show that your method on average would help a histopathologist correctly diagnose patients more often than using a non-Bayesian approach. That should be the goal. In practice, you would need a way to provide at least a proxy measure for this. You might consider an experiment like the one in https://arxiv.org/pdf/1901.09392, where they use a very carefully designed controlled experiment to showcase things. Basically, you need a way to simulate the use-case you care about and show that the best alternative (i.e., non-Bayesian) doesn't solve the task but yours does.

- The above points question the mismatch between the framing of the problem, e.g., "BNN ...evaluate their faithfulness", and the evaluation metrics. Figure 1 shows variances of explanation...but why is that important? Yes, the approach can produce a distribution over explanations. But why is that useful for the use cases? Why would having this information actually change what you do?
   - The evaluation metrics do not match this goal. Merely checking if the explanation overlaps with the object does not actually measure what is needed because the explanation method could game the metric by simply highlighting the foreground object all the time regardless. See e.g., the many papers on adversarial XAI: https://github.com/hbaniecki/adversarial-explainable-ai.  While this might not be true for your method, the metric doesn't seem to measure what is important.

---

> ### Author Response · Authors · 2025-08-13
> **Answer to the Reviewer Na4v**
>
> We sincerely thank the reviewer for their detailed and thought-provoking comments. We appreciate the positive remarks on the clarity of the paper, the illustrative value of the histopathology and watermark examples, and the recognition that understanding and explaining uncertainty is a critical direction for XAI. Below we address the main points of the reviewer.
>
> 1. **Use Cases and Decision-Relevance of Explanation Uncertainty**
> > "The paper… lacks concrete use-cases of when the explanation will actually be useful to someone… What decision could it actually help?"
>
> We agree that clearly connecting explanation uncertainty to end-user decisions is essential, and we appreciate the request to sharpen this connection. Our work does not claim to present a finalized decision-support system, but rather to introduce a tool that supports diagnosis to expose variability in feature attributions due to model uncertainty. This can serve as a building block for a downstream interpretability pipeline, particularly in high-stakes domains such as medicine.
> In the histopathology use case, we do not assume that the end-user is a pathologist, but rather a developer or model auditor interested in understanding when and where the explanation of the model’s prediction is uncertain, even if the model prediction itself appears confident. For example, when the explanation variability is high, it can be used to label samples for human inspection or for training data review. This uncertainty signal is complementary to prediction confidence and can indicate instability in decision rationale, which might otherwise remain hidden.
> We acknowledge that we do not provide a human-in-the-loop evaluation, and we agree that this is a valuable direction for future work. We have clarified the scope of our contribution in the revised manuscript and tempered the claims accordingly. We also expanded the discussion of relevant alternatives, such as explanation-based influence functions, and clarified where our method complements these approaches or differs.
>
> 2. **Evaluation Metric and Alignment with Use Cases**
> > "Why does a better score on this metric imply that your method would perform better on the use case?"
>
> We acknowledge the reviewer’s concern that overlap-based metrics (e.g., object coverage) are limited in capturing practical utility. In this work, we used these metrics to benchmark the aggregated explanations against established XAI outputs, not as a direct proxy for decision-making quality.
> That said, we agree that adversarial robustness and explanation faithfulness are important and are now discussed more explicitly in the limitations section. We clarify that our goal is not to propose new evaluation metrics but to provide a method that reveals previously hidden epistemic variation in explanations, which can be valuable for model debugging and might provide additional model insights.
> We appreciate the reviewer’s request for more grounded use-case discussion, and we believe the added clarifications and examples strengthen the practical motivation of our work. While our framework is not a full decision-support pipeline, we see it as a crucial step toward enabling trustworthy explanations under uncertainty. We believe that the revised manuscript substantially strengthens the presentation of our contribution and its role as a foundation for future use cases, where quantifying explanation uncertainty can contribute to a more faithful and trustworthy model understanding.

---

> > ### Comment · Reviewer_Na4v · 2025-08-15
> >
> > I have read the author's rebuttal. My main concerns remain that you have not explained nor demonstrated the actual utility of the explanation method. I would encourage you to work on concrete use-cases that can convincingly show the utility of your method. Note that these do not necessarily need to be user studies but could be appropriate proxy measures with careful and convincing experimental setup that matches the real-world use case.

---

> > > ### Author Response · Authors · 2025-08-25
> > > **Official Comment to Reviewer Na4v**
> > >
> > > We sincerely thank the reviewer for answering to our rebuttal and for once again stressing the importance of demonstrating the utility. We greatly value this feedback, and it helped us to refine both the tone and scope of our contribution.
> > >
> > > **Utility by quantitative evaluation:**
> > > We fully agree that a convincing demonstration of utility ideally involves use-case aligned experiments. Within the scope of this work, we position our contribution more modestly and show that incorporating epistemic uncertainty into explanations leads to measurable improvements. The established XAI quality metrics, such as pixel flipping could not be used out of the box,  since the decrease in model performance over the perturbation of relevant pixels can not be compared directly regarding the different number of relevant pixels for the intersection, mean and union explanation. Therefore we used AUC and mean attribution (MA) metrics to capture complementary desiderata: AUC reflects sensitivity by measuring how well highlighted features support the predictions, while MA reflects stability by quantifying the consistency of attributions across the explanations. Our results suggest that uncertainty-aware explanations can be adapted to different practical needs, i.e., the union explanations improve AUC, reflecting higher sensitivity by capturing a broader set of potentially relevant features, whereas the intersection explanations improve MA, reflecting higher stability by highlighting only the most consistent features.
> > >
> > > Together, these results illustrate how uncertainty-aware explanations can be tuned toward different practical needs. While these are not user studies, they constitute carefully chosen proxy measures that map onto realistic decision goals such as avoiding missed evidence (sensitivity) or guarding against spurious cues (stability).
> > >
> > > **Clustering as a qualitative lens:**
> > > In addition, clustering provides complementary qualitative insight by organizing posterior explanations into diverse patterns. We do not present this as a final diagnostic tool, but as a visualization that might support practitioners to explore if a model relies on heterogeneous or potentially unstable reasoning.
> > > Future work:
> > > We agree wholeheartedly with the reviewer that more concrete use-case evaluations are the next step. In the revision, we highlight possible directions—for example, in pathology, Bayesian XAI may consistently reveal both spurious and real evidence as separate clusters, thereby reducing the risk that users overlook true signals when spurious cues are present. While such deployment-oriented evaluations are beyond the current scope, our framework provides a foundation on which future work can be built.
> > > We sincerely hope the reviewer will recognize our contribution as a piece of fundamental research that lays the foundation for future work, providing both theoretical insights and practical tools to explore the usefulness in real-world applications.
> > >
> > > **Revised manuscript:**
> > > In light of the reviewer’s feedback, we have carefully revised the manuscript to:
> > > 1. Present our contribution as a framework for producing and analyzing distributions of explanations under model uncertainty.
> > > 2. Emphasize that our empirical results demonstrate improvements on established XAI criteria as proxy measures, not as final decision-support validation.
> > > 3. Clearly mark application-specific evaluations and user studies as future work.
> > > We hope that these clarifications address the reviewer’s concerns: our work should be read not as a complete solution to decision-making utility, but as a principled step that provides both quantitative and qualitative tools for exploring explanation uncertainty, with the potential to be extended into full use-case evaluations in future research.

---

### Review · Reviewer_A7do · 2025-08-01

**Summary Of Contributions:**

The paper introduces a new framework called Uncertainty-Aware Explanations (UAE) tailored for Bayesian Neural Networks (BNNs). It addresses the challenge of quantifying and visualizing the uncertainty inherent in model explanations, a facet largely overlooked by traditional explainability methods primarily designed for deterministic models. Basically, the idea is novel and the paper is well-structured. The theoretical depth is also good. However, the experiments might be improved.

**Audience:**

Yes

**Claims And Evidence:**

Yes

**Requested Changes:**

There is no time complexity analysis both theoretically and empirically. Give more discusson on them.

The authors only include metrics AUCROC and MA are used in the exp, however, for xAI, some more popular metrics such as fidelity. For example, the following papers are discussing it with fidelity:
A benchmark for interpretability methods in deep neural networks. NeurIPS, 2019
F-FIDELITY: A ROBUST FRAMEWORK FOR FAITHFULNESS EVALUATION OF EXPLAINABLE AI. ICLR, 2025.
Please add more results on fidelity, ROAR, F-Fidelity for comparison if possible, or explain why these are not suitable.

**Strengths And Weaknesses:**

Pros:
1. The framework incorporates uncertainty estimates into explanations, providing a more nuanced understanding of feature relevance and model confidence, which is particularly valuable in safety-critical domains.

2. The method leverages the probabilistic nature of Bayesian Neural Networks, allowing explanations to reflect the inherent epistemic uncertainty in model predictions and feature attributions.

3. Extensive experiments on synthetic, natural, and medical datasets demonstrate versatility and practical relevance, showing the method's robustness in various contexts.

Cons:
1. Sampling from the posterior distribution to generate multiple explanations increases computational cost compared to single-point explanation methods, which may impede real-time applications.

2. There is no time complexity analysis both theoretically and empirically.

3. The authors only include metrics AUCROC and MA are used in the exp, however, for xAI, some more popular metrics such as fidelity. For example, the following papers are discussing it with fidelity:
A benchmark for interpretability methods in deep neural networks. NeurIPS, 2019
F-FIDELITY: A ROBUST FRAMEWORK FOR FAITHFULNESS EVALUATION OF EXPLAINABLE AI. ICLR, 2025.

---

> ### Author Response · Authors · 2025-08-13
> **Answer to the Reviewer A7do**
>
> We thank the reviewer very much for their constructive feedback and their appreciation of the theoretical depth, the integration of uncertainty into feature relevance, and the robustness of our experiments. Below we address the reviewers' suggestions in detail.
>
>
> 1. >“There is no time complexity analysis both theoretically and empirically. Give more discussion on them.”
>
> We agree that the computational cost is an important consideration. Our proposed method scales linearly with the number of posterior samples $N$, as it only requires computing $N$ standard attribution maps followed by aggregation.
> To improve the clarity for the reader, we have revised the manuscript as follows, by adding a "Computational Complexity" section 4.4
>
> The actual cost varies by posterior approximation method. For example, in dropout-based BNNs [Gal and Ghahramani, 2016], inference differs only slightly from deterministic models, adding minimal cost via Bernoulli sampling and element-wise masking. More expensive methods like HMC are compatible but outside our empirical scope. A detailed time complexity comparison across BNN variants is beyond this paper’s focus, but we note that our explanation method is designed to be agnostic to the specific BNN formulation used, and compatible with scalable approximations.
>
> 2. > “The authors only include metrics AUCROC and MA are used in the exp, however, for xAI, some more popular metrics such as fidelity... Please add more results on fidelity, ROAR, F-Fidelity... or explain why these are not suitable.”
>
> We appreciate the pointer to relevant literature. In this work, we focused on evaluating how existing XAI methods behave under posterior sampling with a focus on quantifying and analysing epistemic variability. Therefore, we used AUC-ROC and mAP as metrics, which are applicable in the probabilistic setting. Our primary contribution is in aggregating and analyzing epistemic variability, rather than introducing a new XAI method. While we agree that metrics like Fidelity, ROAR, and F-Fidelity are valuable, they are typically developed for deterministic models with single explanations. Their extension to probabilistic models and ensembles of explanations is non-trivial and not yet standardized. Developing such metrics for the probabilistic case is an exciting direction for future work, and we plan to explore this.
>
> Bibliography:
> [Gal and Ghahramani, 2016]  Gal, Yarin, and Zoubin Ghahramani. "Dropout as a bayesian approximation: Representing mo

---

### Review · Reviewer_eXSm · 2025-08-05

**Summary Of Contributions:**

This paper proposes an explainable AI (XAI) method for Bayesian Neural Networks (BNNs), building on existing XAI techniques for Deep Neural Networks (DNNs). The authors aim to quantify the uncertainty in model predictions. They evaluate their method using datasets from ImageNet and cancer research, demonstrating that their approach works efficiently.

**Audience:**

No

**Claims And Evidence:**

Yes

**Requested Changes:**

Computational Cost:
   While the model demonstrates strong performance, its computational cost is notably high. I recommend exploring strategies to reduce the expense associated with sampling from the posterior distribution, such as approximation techniques or more efficient sampling methods, to make the approach more feasible for large-scale applications.

Explanation Diversity:
   The observed diversity in the explanations might stem from local perturbations around a single mode, which may not truly reflect distinct predictive strategies. It would be beneficial to further analyze whether these variations are genuinely due to different predictive strategies or simply artifacts of the perturbation process. A clearer explanation of how the model handles multiple modes in the data could strengthen this aspect of the paper.

Evaluation Metrics:
   I suggest incorporating more standard evaluation metrics for XAI, such as Fidelity, Stability, or Interpretability, alongside the AUC-ROC and Mean Average Precision (MAP). Using these additional metrics would provide a more comprehensive assessment of the model's explanatory power and could make the findings more robust.

**Strengths And Weaknesses:**

Strengths:

The paper is well-written and clearly presents the idea.

The concept is relatively novel and offers useful insights for the research community.

The experimental results support the proposed approach.

Weaknesses:

While the model performs well, its computational cost is high. Sampling from the posterior distribution leads to significant expense.

The variations in the explanations observed could be due to local disturbances around a single mode, rather than indicating fundamentally different predictive approaches.

I believe it would be better to employ more standard metrics for evaluating XAI, instead of relying solely on AUC-ROC and Mean Average Precision (MAP)

---

> ### Author Response · Authors · 2025-08-13
> **Answer to the Reviewer eXSM**
>
> We sincerely thank the reviewer for their thoughtful and constructive feedback. We are particularly grateful for the positive remarks on the novelty, clarity, and practical relevance of our work. Below, we address the key concerns raised in the review.
>
> 1. **Computational cost of posterior sampling**
> > "Sampling from the posterior distribution leads to significant expense."
>
> We agree that posterior sampling introduces computational overhead compared to single forward passes in deterministic networks. However, we would like to emphasise that the computational cost of our proposed explanation aggregation scheme grows only linearly with the number of sampled models, as it only requires computing standard attribution maps over each of the $N$ samples, followed by a simple aggregation step.
> To improve the clarity for the reader, we have revised the manuscript by adding "Computational Complexity" section 4.4.
>
> Regarding posterior approximation, the computational complexity largely depends on the specific BNN approximation used. For instance, in variational dropout-based models (e.g., [Gal and Ghahramani, 2016]), inference differs only marginally from deterministic models, requiring a single additional Bernoulli mask sampling and an element-wise multiplication per layer. These lightweight operations incur minimal cost and remain efficient at inference time.
> While some posterior approximation methods can be more expensive, the choice of the approximation technique is orthogonal to our contribution.  A detailed exploration of efficiency trade-offs across BNN methods is beyond the scope of this paper. Our method is compatible with most common BNN approximations, and we expect future work on more efficient approximations to further reduce the computational overhead.
>
> 2. **Explanation Diversity may arise from local perturbations**
> > "The variations in the explanations observed could be due to local disturbances around a single mode, rather than indicating fundamentally different predictive approaches."
>
> We appreciate this insightful comment. Indeed, one cannot assume that explanation diversity automatically implies distinct predictive strategies unless the explanation method used is faithful and robust. In our work, we rely on commonly accepted and locally faithful explanation methods (e.g., Integrated Gradients and LRP), and our analysis in the FMNIST experiments (Fig. 3) demonstrates that explanations can be clustered into qualitatively distinct groups that correspond to perceptually different attribution patterns. This suggests that even if the sampled models originate from the same approximate mode, the epistemic uncertainty in BNNs can still manifest in diverse explanation structures.
> We agree with the reviewer that a more in-depth theoretical investigation of whether this diversity stems from multiple posterior modes or local variance around a dominant mode would be valuable future work. Nonetheless, our empirical results suggest that the observed variation is visually and semantically meaningful, aligning with the intended purpose of our proposed explanation framework.
>
> 3. **Suggestion to use additional XAI metrics (Fidelity, Stability, Interpretability)**
> > "It would be better to employ more standard metrics for evaluating XAI, instead of relying solely on AUC-ROC and MAP."
>
> Thank you for this suggestion of extending the XAI evaluation. In this work, our goal was to evaluate our aggregation strategy over existing XAI methods, rather than to introduce a new XAI method per se. We therefore used metrics (AUC-ROC, mAP) that allow for evaluating the probabilistic models. While we agree that XAI evaluation metrics such as Fidelity and Stability are important metrics for measuring explanation quality, they have not yet been adapted or standardized for probabilistic models and multi-explanation outputs, such as the ones produced by our method. Defining such metrics in a probabilistic setting is an important direction for future work that will significantly contribute to the XAI evaluation field.
>
>
> Bibliography:
> [Gal and Ghahramani, 2016]  Gal, Yarin, and Zoubin Ghahramani. "Dropout as a bayesian approximation: Representing model uncertainty in deep learning." international conference on machine learning. PMLR, 2016.

---

> > ### Comment · Reviewer_eXSm · 2025-08-26
> >
> > Thank you for your effort to address my concerns. I am happy with this version being accepted.

---

### Author Response · Authors · 2025-08-12
**Official Comment**

We sincerely thank all reviewers — eXSm, A7do, Na4v, ovyt, and noQ2 — for their careful evaluation, thoughtful comments, and constructive feedback, which enabled us to significantly improve and strengthen our work. We deeply appreciate the time and insight you have devoted to reviewing our submission.

We note with gratitude several positive points that emerged across the reviews:

- **Clarity and writing quality**. Recognized as well-written and well-organized by eXSm, noQ2, and Na4v.
- **Conceptual value / novelty of uncertainty-aware explanations for BNNs**. Called relatively novel and useful by eXSm and A7do; ovyt affirmed the topic’s importance for the audience while requesting clearer differentiation from prior work.
- **Method-agnostic and broadly compatible framework**. Highlighted as a strength by noQ2 (works with multiple attribution methods and BNN inference schemes) and echoed in ovyt (generality) and Na4v (applicable to standard DNNs with dropout).
- **Clustering as an intuitive way to explore predictive diversity**. Explicitly praised by noQ2 as an effective and intuitive mechanism to explore diversity of decision rationales.
- **Experimental breadth / practical relevance**. Appreciation for coverage across synthetic, natural, and medical data by A7do; broad applicability across inference methods and datasets by noQ2; concrete examples (histopathology, watermark) valued by Na4v.
- **Soundness / theoretical depth**. Theoretical depth noted by A7do; mathematical correctness and justified claims by ovyt.

In the revised manuscript, we have incorporated the requested changes, highlighted the main revisions in green for ease of verification, and fixed several grammatical and stylistic issues throughout. We revised the manuscript to clarify the scope and tightened claims: specifically, we softened language around “revealing multimodality,” now framing our contribution as surfacing explanation diversity under an approximate posterior (Abstract, Introduction, “Exploring multi-modality,” Discussion); added a concise computational complexity paragraph showing linear overhead. We refined the Evaluation section to acknowledge Fidelity/ROAR/F-Fidelity, explain why direct use is non-trivial for distributions of explanations. We further refined the Conclusion section and added a clear Limitations paragraph.

We now address each review individually below, providing responses to the main points of the reviewers.

---

### Decision · Action_Editor_1osH · 2025-08-18

**Recommendation:** Accept with minor revision

**Additional Comments:**

The authors are suggested to clarify the novelty of their work in comparison to the four related papers mentioned by Reviewer ovyt. Additionally, providing real-world use cases would help demonstrate the practical value of the proposed method.

**Audience:**

Yes

**Audience Explanation:**

This paper introduces a new uncertainty-aware explanation for Bayesian Neural Networks (BNNs). The approach quantifies and visualizes the uncertainty inherent in model explanations. The proposed method is relatively novel, and the paper is well written. The uncertainty-aware explanation is of important value in the field of explainable AI, and experts in diverse directions would be interested in the new method in this paper. The authors are suggested to better clarify the application utility of the proposed method. More concrete application cases and more detailed analysis are needed. Although compared to other AI directions, explainable AI prioritizes theoretical advances over applications, showing practical value could significantly increase the work's influence.

**Claims And Evidence:**

Yes

**Claims Explanation:**

This paper introduces a new uncertainty-aware explanation for Bayesian Neural Networks (BNNs). The approach quantifies and visualizes the uncertainty inherent in model explanations. The proposed method is grounded in mathematical theory, and different experiments have demonstrated the explanation power.